# Benchmarking common quantification strategies for large-scale phosphoproteomics

Alexander Hogrebe [1], Louise von Stechow[1], Dorte B. Bekker-Jensen[1], Brian T. Weinert[1], Christian D. Kelstrup [1] & Jesper V. Olsen [1]

Comprehensive mass spectrometry (MS)-based proteomics is now feasible, but reproducible quantification remains challenging, especially for post-translational modifications such as phosphorylation. Here, we compare the most popular quantification techniques for global phosphoproteomics: label-free quantification (LFQ), stable isotope labeling by amino acids in cell culture (SILAC) and $MS^2$- and $MS^3$-measured tandem mass tags (TMT). In a mixed species comparison with fixed phosphopeptide ratios, we find LFQ and SILAC to be the most accurate techniques. $MS^2$-based TMT yields the highest precision but lowest accuracy due to ratio compression, which $MS^3$-based TMT can partly rescue. However, $MS^2$-based TMT outperforms $MS^3$-based TMT when analyzing phosphoproteome changes in the DNA damage response, since its higher precision and larger identification numbers allow detection of a greater number of significantly regulated phosphopeptides. Finally, we utilize the TMT multiplexing capabilities to develop an algorithm for determining phosphorylation site stoichiometry, showing that such applications benefit from the high accuracy of $MS^3$-based TMT.

[1] Novo Nordisk Foundation Center for Protein Research, Proteomics Program, Faculty of Health and Medical Sciences, University of Copenhagen, Blegdamsvej 3b, 2200 Copenhagen, Denmark. Correspondence and requests for materials should be addressed to J.V.O. (email: jesper.olsen@cpr.ku.dk)

Nanoflow liquid chromatography tandem mass spectrometry (LC-MS/MS)-based quantitative phosphoproteomics technology has revolutionized cell biology in the past decade[1–3]. This has mainly been driven by advances in MS instrumentation[4–7], phosphopeptide enrichment strategies[8,9], peptide chromatography[10–12], and computational proteomics tools[13,14]. The basis for any biological interpretation of phosphoproteomics data is the quantification of the identified phosphopeptides. This can be done either relatively by calculation of ratios between conditions, or absolute within conditions, often referred to as phosphorylation site stoichiometry or occupancy[15–17]. Stable isotope labeling by amino acids in cell culture (SILAC) has long been the preferred method for phosphopeptide quantification[1,18], but recently label-free quantification (LFQ)[2,19,20] and isobaric tandem mass tags (iTRAQ, TMT)[3] have become popular for phosphoproteomics.

SILAC is a full (MS[1]) scan-based quantification method. Stable heavy isotope-labeled or unlabeled amino acids are incorporated metabolically into cells, and differentially labeled cell populations can be mixed directly after lysis[21]. SILAC is arguably the most accurate quantification technique. However, SILAC is limited to cell lines and a maximum of three conditions per sample in routine analysis. In addition, SILAC and other labeling approaches with MS[1]-based quantification have inherently higher MS[1] spectral complexity. The resulting over-sequencing of peptide isotope-variants leads to a decrease in total peptide identifications[22,23]. LFQ does not require incorporation of stable isotopes, but instead relies on comparison of peptide MS[1] signal intensities between MS runs and is therefore easy to integrate in most experimental workflows[24]. Conversely, LFQ requires each sample to be measured individually and often suffers from "missing quantification values" between them. To circumvent this, isobaric labeling techniques such as iTRAQ[25] and TMT[26] were developed, which enable the simultaneous measurement of up to 11 samples labeled on peptide-level[27]. This approach is known as sample multiplexing and allows highly sophisticated biological study designs. As a tradeoff compared to LFQ, isobaric quantification approaches usually yield decreased peptide identification rates[22,28] and lowered accuracy when quantified on MS[2]-level[23]. The latter is caused by ratio compression via impure MS[1] precursor isolation[29]. This leads to ratios between conditions converging towards the median value, i.e. usually towards unity[30], but also higher apparent precision[31]. To circumvent the ratio compression issue, MS[3]-based approaches have been introduced, which aim at isolating target fragment ions from MS[2] and quantifying the target reporter ions separately[32,33]. However, these and similar approaches to prevent ratio compression are usually either limited to specialized MS instruments or not routinely usable yet[34–37].

Different quantification approaches have been compared in a systematic way for protein quantification, but not yet for phosphoproteomics, where sampling of several peptides for each protein ratio is not possible[22,23,31,38–40]. Furthermore, the impact of individual technical parameters for actual biological findings, such as quantification precision and accuracy for deeming phosphopeptides significantly regulated or estimation of fractional phosphorylation site stoichiometry, has not been assessed previously[41].

Here, we systematically compared phosphoproteomics quantification approaches with a focus on technical parameters and their performance in biological studies. We tested the four most common quantification strategies for phosphoproteomics: LFQ, SILAC, and MS[2]- and MS[3]-based TMT. Initially, we assessed the extent of ratio compression in different phospho-optimized MS[2]- and MS[3]-based TMT methods, and compared their quantification accuracy and precision to LFQ and SILAC using a controlled phosphoproteome mixture with defined ratios. We then tested how these findings translate into identification of significantly regulated phosphopeptides in the well-studied DNA damage response (DDR)[18] under instrument-time-limiting conditions. Finally, we evaluated the capability of TMT to determine fractional phosphorylation site stoichiometry in large-scale phosphoproteomics data sets. For this purpose, we developed a 3D multiple regression model (3DMM)-approach that makes use of the high TMT multiplexing capabilities. Our results indicate that even with ratio compression, MS[2]-based TMT is best suited for the quantification of complex biological phosphoproteomics samples, while the high accuracy of MS[3]-based TMT quantification is ideal for calculation of phosphorylation site stoichiometry.

## Results

**MS[3]-based TMT enables best ratio decompression.** In this analysis, we aimed at evaluating different MS quantification methods both from a technical perspective, as well as with a biological focus on identification of significantly regulated phosphopeptides. Since the latter is influenced by technical parameters such as quantification accuracy, we first assessed to which extent ratio compression affecting the accuracy of MS[2]-based TMT measurement can be decompressed using MS[3]. To measure actual ratio-compressed MS[2]-quantified ratios, we treated U2OS human osteosarcoma cells for 2 h with doxorubicin (DOX), a potent genotoxic agent inducing a global phospho-signaling response, or DMSO as a control (C). TiO$_2$-enriched tryptic phosphopeptides were then measured on an Orbitrap Fusion Lumos instrument by MS[2]- and four different phospho-optimized MS[3]-based approaches (Fig. 1a, Supplementary Table 1). The MS[3]-based methods either employed MS[2] analysis in the orbitrap (OT) or ion trap (IT), and used different settings for MS[2]-fragmentation and MS[3]-ion selection. Three of them used multi-stage activation (MSA)-combined low energy collisional-induced dissociation (CID) and synchronous precursor selection (SPS) of the ten most abundant MS[2] fragment ions[33]. The fourth method applied CID without MSA followed by neutral loss (NL)-triggered phospho-peak selection for MS[3], as previously reported by Erickson et al.[39]. All raw files were then processed by the MaxQuant software suite (www.maxquant.org) for identification and quantification[13].

To evaluate MS[3]-based decompression of MS[2]-measured DOX vs. C ratios, we performed a linear correlation of the 5% most up- and downregulated phosphopeptide ratios after log2-transformation and used the slope as a readout. As expected, all MS[3]-based TMT methods were able to decompress the observed MS[2]-based TMT ratios, but to different extents (Fig. 1b, Supplementary Figs. 1 and 5g). The SPS-MS[3]-based method with MSA-CID and ion trap-MS[2] analysis performed worst, only decompressing log2 ratios by a factor of 1.35. The NL-triggered method showed a better decompression of 1.72, but at the same time generated the lowest number of phosphopeptide ratios of all five methods (Fig. 1c). The low numbers of phosphopeptide identifications for the two ion trap MS[2]-based methods indicated that phosphopeptide-identifications benefited significantly from MS[2]-analysis in the orbitrap. The best performing method was termed MS[3] orbitrap multiple charge state (OT MC) and yielded a log2 ratio decompression of 1.96 and the highest MS[3]-based number of ratio quantifications. It combined phospho-optimized MSA-CID-MS[2]-based orbitrap analysis with SPS-MS[3] and an ion selection width, which was inversely correlated with the precursor charge state. We further noted that all MS[3]-based analyses had a significantly lower number of phosphopeptide ratios and lower median signal-to-noise ratios of the TMT reporter ions (Fig. 1d) than the MS[2]-based approach.

**MS²-based TMT is the most precise quantification method.** The ability of a quantification method to identify phosphorylation sites as significantly regulated depends on different technical parameters. Thus, after confirming that MS³-based TMT methods can decompress MS²-measured phosphopeptide ratios to different extents, we next compared the quantification accuracy and precision of these approaches to that of LFQ and SILAC. For this purpose, we used a mixed species approach in which we diluted phosphopeptides enriched from yeast at fixed 1:4:10 ratios into a 1:1:1 background of HeLa phosphopeptides. This way we can assess how the different methods would quantify the expected

ratios (Fig. 2a). Since SILAC labeling of wild-type yeast cells is limited to lysine, we digested proteins only with endoproteinase Lys-C for all methods in this experiment[42]. Importantly, all LFQ and TMT-labeled samples used for the dilution experiments were generated from the same lysate, but this was not possible for SILAC, which was heavy stable isotope-labeled already during cell culture. Thus, it should be noted that while all LFQ and TMT samples were essentially the same peptides mixed in different abundances, the SILAC ratios might be influenced by unavoidable biological sample-variation, which we expect to have an impact on assessing its precision. In MaxQuant, the LFQ-measured samples were processed both with and without activation of the MaxQuant feature match-between-runs (MBR), and SILAC-measured samples with and without MBR and the requantify (REQ) feature. MBR is a method for transferring identification information between samples, leading to increased peptide identifications and fewer missing values. MBR is now widely employed in LFQ- and SILAC-based studies and similar concepts are implemented in many proteomics software tools.

To achieve reliable biological interpretation, quantification methods have to be both accurate and precise. To assess this, we first compared how accurate the different quantification methods could measure the 4:1 and 10:1 yeast phosphopeptide ratios (Fig. 2b, Supplementary Fig. 2). LFQ and LFQ MBR both turned out to be the most accurate methods, slightly overestimating ratios on median. SILAC was almost as accurate as LFQ, but underestimated ratios on median. However, the quantification accuracy drops significantly when also activating the REQ feature together with MBR, indicating that REQ trades accuracy for an increase in number of quantified sites and thus should be used with caution (Fig. 2b). As expected, MS²-based TMT ratios were heavily affected by ratio compression, resulting in the lowest accuracy of all compared quantification methods. MS³-based TMT methods can rescue the compression to different extents, with the TMT MS³ OT MC method being closest to SILAC. Since the true target ratios were known, we were able to calculate the fraction of reporter ion intensity coming from contaminating ions for MS²-based TMT. We found that it negatively correlated with MS¹ precursor intensity, Andromeda MS/MS score and precursor isolation fraction (PIF) (Supplementary Fig. 3). However, based on the Pearson correlation coefficients, none of these turned out to be a robust predictor, which is consistent with what was previously reported for the PIF value[23]. We next calculated the mean squared errors (MSE) as the sum of positive bias and variance for each method. These represent the quantification error in accuracy and precision, respectively, and thus allowed for a direct comparison of these two parameters (Fig. 2c, Supplementary Fig. 2d). Of all methods tested, MS²-based TMT yields the highest precision but lowest accuracy. Furthermore, the

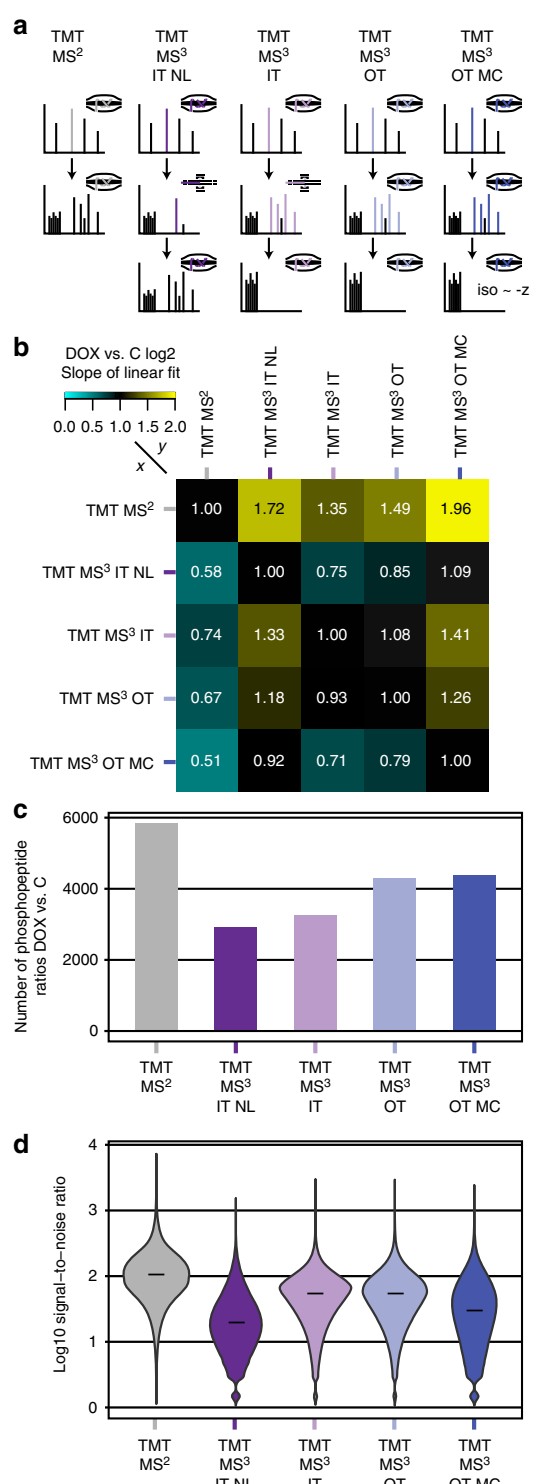

**Fig. 1** Evaluation of phosphorylation-optimized MS²- and MS³-based TMT methods. **a** Colored peaks illustrate MSⁿ peak selection. MS² analysis either took place in the orbitrap (OT) or ion trap (IT). Ion selection for MS³ analysis was based on synchronous precursor selection (SPS) or neutral loss (NL)-triggered peak isolation. In the multiple charge state (MC) method, the MS³ isolation width was decreased for higher charge states. IT, OT and OT MC used multi-stage activation (MSA) with neutral loss mass 97.9673 Da. **b** Heatmap of correlation slopes of the 5% highest and lowest log2 ratios for all replicates. U2OS cells were treated 2 h with 5 μM doxorubicin (DOX) or DMSO (C). The resulting TMT sample was measured on an Orbitrap Fusion Lumos three times as technical replicates with each quantification method. **c** Bar plot showing the total number of quantified phosphopeptide DOX vs. C ratios per method for all replicates. **d** Violin plot showing log 10 signal-to-noise ratio distributions of the TMT reporter ions with the median marked as a dash

higher accuracy of MS³-based TMT methods seem to come at the cost of lower precision compared to MS²-based TMT. LFQ, LFQ-MBR and SILAC-MBR-REQ show the lowest precision of all quantification methods. Furthermore, increasing ratios from 4:1 to 10:1 leads to a decrease in precision for all quantification methods.

Next we assessed how the different tradeoffs in accuracy and precision for the quantification approaches would influence their ability to identify phosphopeptides as significantly regulated. This seemed especially interesting for MS²-based TMT, where the apparent increase in quantification precision is easiest explained by the contamination of the TMT reporter ion signal with co-isolated, non-regulated peptides. We would expect that if the gain in precision is indeed caused solely by such a quantification artifact, MS²-based TMT should perform much worse than MS³-based TMT at deeming phosphopeptides significantly regulated. To test this, we analyzed our phosphoproteomics data sets for significantly regulated sites by applying a well-established statistical technique for large-scale omics data, significance analysis of microarrays (SAM)-testing, which uses t-testing with an added background variance parameter termed s0[43]. This s0 parameter gives greater weight to extreme fold changes and should be adjusted to the data set at hand[44]. We used an R package provided by Tibshirani et al.[45], which can automatically estimate optimal s0 based on the tested data, and calculates a d-score representing the degree of significant regulation of each tested phosphopeptide. Since we know that all yeast phosphopeptides should by definition be regulated within our data set, we can use the d-score to calculate true-positive-rates (TPR) and false-positive-rates (FPR) of the upregulated phosphopeptides for each of the quantification approaches, and plot them against each other in a receiver operating characteristic (ROC) curve (Fig. 2d). In such a ROC curve plot, an ideal quantification approach would reach a TPR of 100%, which means all true positive hits were identified as positive, before the FPR becomes greater than 0%. When looking at the 4:1 ratios, we see that LFQ shows the steepest TPR increase, followed by SILAC and MS³ OT MC-based TMT. As expected, MS²-based TMT performs poorly, indicating that its higher apparent precision is indeed not providing robust quantification of low peptide ratios. At the higher 10:1 ratios however, MS²-based TMT performs equally well as MS³ OT MC-based TMT, and even outperforms it at higher FPRs. We would like to stress that this analysis of course depends on many factors, including chosen ratios, total MS ion intensities and the applied statistical test. It is additionally complicating that the number of

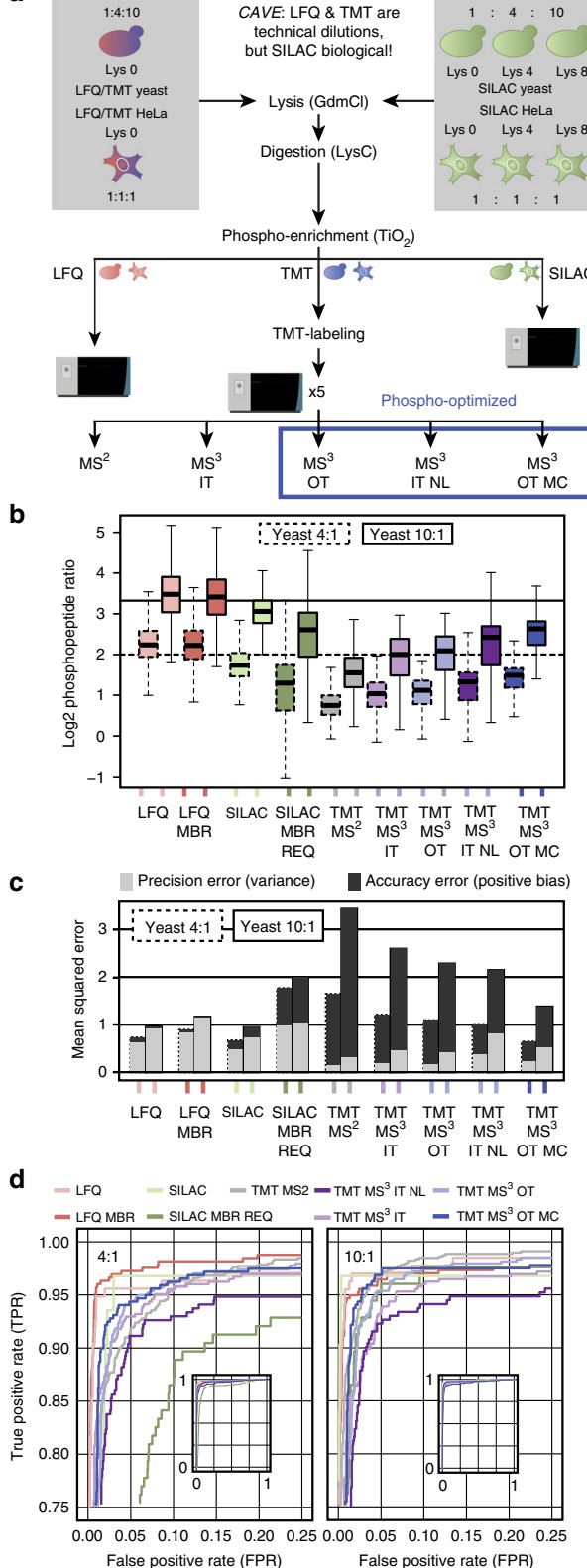

**Fig. 2** Evaluation of quantification methods with focus on accuracy and precision. **a** Yeast phosphopeptides were diluted in fixed ratios 1:4:10 and added to a background of 1:1:1 HeLa phosphopeptides. Same total protein starting amounts were used for each method and SILAC ratios were mixed before digestion. All samples were measured on an Orbitrap Fusion Lumos three times as technical replicates with each quantification method. For SILAC and TMT, MS samples were diluted to contain a total peptide amount equal to one LFQ injection based on protein starting amount. For TMT, all mixing replicates were measured within the same TMT10-plex run. **b** Box plot showing yeast 4:1 and 10:1 phosphopeptide ratios for the different quantification methods and all replicates. Boxes mark the first and third quartile, with the median highlighted as dash, and whiskers marking the minimum/maximum value within 1.5 interquartile range. Outliers are not shown. Both LFQ and SILAC were tested with and without the MaxQuant feature match-between-runs (MBR), and SILAC additionally with both MBR and requantify (REQ) activated. As SILAC-MBR only results were essentially identical to SILAC only, they are not shown here. **c** Mean squared errors were calculated as a sum of positive bias and variance for each method and all replicates. **d** Receiver operating characteristic (ROC) curves were calculated by using the d-score from SAM testing as an indicator for significant regulation at 4:1 and 10:1 dilution. SAM testing for significantly regulated phosphopeptides was performed at default settings (s0 estimation automatic). ROC plots are presented as zoomed-in excerpts from the total plots, shown on the lower right each

identified yeast and human peptides vary between the quantification approaches due to the inherent stochastic behavior of the data-dependent acquisition (DDA). This is especially true for LFQ/TMT vs. SILAC since they are essentially different biological samples (Supplementary Table 2). Nevertheless, we can conclude that, especially for the biologically more interesting 10:1 ratios, LFQ, SILAC, MS³ OT MC-based and even ratio-compressed MS²-based TMT, all seem to provide good and comparable compromises between sensitivity and specificity.

**MS²-based TMT excels at identifying significant regulation.** After comparing the different quantification methods in a technical setup, we next evaluated if the increased accuracy of MS³-based TMT over MS²-based TMT translates into an advantage in a biologically relevant setting. For this purpose, we used the well-studied DDR as a model system (Fig. 3a). Cells were treated with genotoxic agents doxorubicin (DOX), 4-nitroquinoline 1-oxide (4NQO) or DMSO as a control (C) in biological triplicates for 2 h before lysis. To provide a comparison that takes into account time-limiting conditions of deep phosphoproteomics

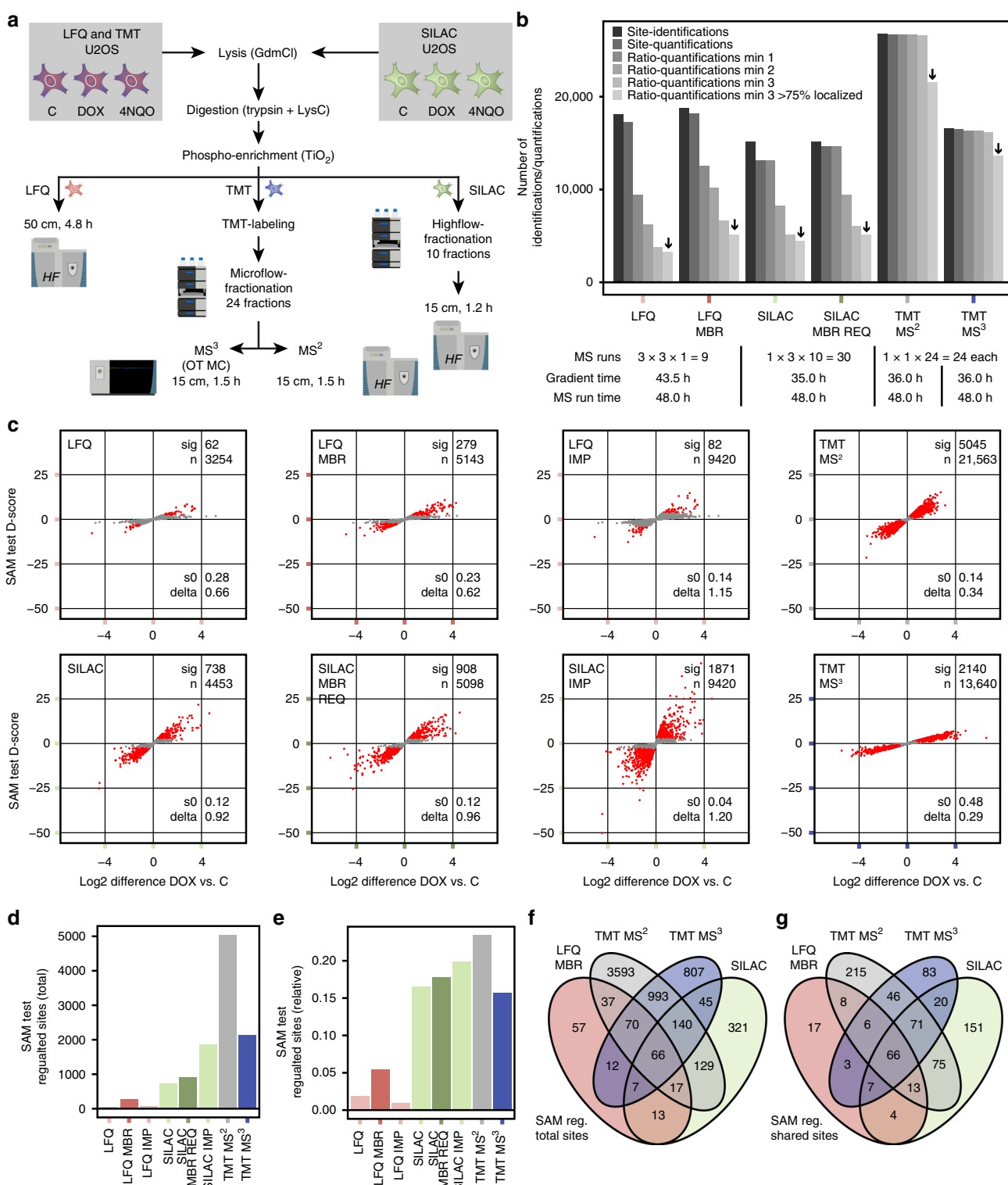

experiments, each method was limited to 2 days of MS instrument time. The nine LFQ samples were analyzed as single shot ~5 h LC-MS/MS runs. The three SILAC replica and the TMT samples were fractionated offline by high-pH reversed phase chromatography[11,12] into ten and 24 fractions, and analyzed by 70- and 90-min LC-MS/MS gradients, respectively.

We first looked at the numbers of identified and quantified phosphorylation sites for each of the four quantification methods, with 35,587 sites identified in total (Fig. 3b). As expected, LFQ with 18,057 phosphorylation sites identified more sites than SILAC with 15,119, due to the latter's increase in $MS^1$ spectra complexity and $MS^2$ fragmentation of redundant SILAC peptide variants. $MS^2$-based TMT identified 26,784 phosphorylation sites, which is >60% more compared to $MS^3$-based TMT with 16,565 sites. Interestingly, SILAC yields the highest number of phosphorylation site identifications without quantitative information, but this can to some extent be rescued by activating the REQ feature. When calculating ratios between doxorubicin and control and requiring values for them in all three biological replicates, numbers for LFQ dropped by almost 80%. This significant loss is due to the aforementioned problem of missing values caused by the stochastic nature of DDA and can be reduced by activating MBR. SILAC is impacted by this as well, when requiring quantification in more than one biological replicate. Importantly, both $MS^2$- and $MS^3$-based TMT are essentially unaffected by missing values, since all conditions were measured within the same TMT10-plex sample.

To assess the biological effect of 2 h doxorubicin treatment on cellular signaling, we next aimed at identifying significantly up- or downregulated phosphorylation sites. For this purpose, we only used phosphorylation site ratios quantified in all three biological replicates and with an at least 75% probability of correct phosphorylation site localization (Fig. 3b). These strict requirements exclude most falsely quantified or localized sites, which is commonly done in cell signaling studies. We then identified significantly regulated phosphorylation site ratios among the remaining sites via SAM testing (Fig. 3c). Due to its low precision and phosphopeptide coverage, LFQ yielded only 62 significantly regulated phosphorylation sites, and even using nearest neighbor-based imputation of values only slightly increased this number (Fig. 3d). Activating MBR on the other hand multiplied the number of regulated sites more than four times to 279. SILAC performs significantly better with 738 sites deemed significantly regulated, and 908 when activating MBR and REQ. Interestingly, this number doubles when using imputation to account for missing values. Still, the TMT-based methods identify the highest total numbers of significantly regulated sites, with 2140 for the $MS^3$-based and 5045 for the $MS^2$-based approach. The more than twofold increase in hits for $MS^2$-based over $MS^3$-based TMT

seems to be mainly caused by the higher total number of phosphopeptide identifications for $MS^2$-based TMT. In addition, the increase in accuracy of $MS^3$-based over $MS^2$-based for TMT quantification is essentially negated by its even bigger decrease in precision. This is shown when looking at relative numbers, as $MS^2$-based TMT is able to deem a larger fraction of its quantified phosphorylation sites as significantly regulated than the $MS^3$-based approach (Fig. 3e). Despite these substantial differences in total numbers, at least 57% of phosphorylation sites tested in all methods were identified as significant in at least one other method as well (Fig. 3f, g).

We would like to stress that the actual number of significantly regulated sites identified varies significantly depending on which data normalization approach and statistical test are used (Supplementary Table 3). Especially for LFQ, the number of hits can vary from 0 to 2146. Importantly however, in all the applied tests, the relative conclusions from above do not change, with LFQ always identifying the least number of significantly regulated phosphorylation sites. We were speculating that this could be influenced by the measurement of LFQ in single shots on a long 50-cm column on a very long 290-min gradient. To test this, we repeated the experiment only for LFQ with DOX treatment, but this time varied the gradient length to include 30, 90 and 180 min on a 15-cm column, and our original 290-min 50-cm approach (Fig. 4a). This new data was measured on a Q Exactive HF-X instead of the HF used in our original data set, which leads to slightly better peptide quantifications for LFQ (Fig. 4b). This might be influenced by the higher scan speed and brighter ion source of the HF-X, which could lead to higher ion intensities and thus better statistics, even though we have not tested this in a direct comparison[46]. Importantly, we still achieve the highest total number of significantly regulated phosphorylation sites with a 290-min 50-cm approach, even though the increase in regulated sites is not correlating well with the high increase in LC-MS time as compared to the very short 30-min gradient (Fig. 4c, d). We found that a much more efficient approach to boost the numbers of significantly regulated phosphorylation sites for LFQ is to measure more than three replicates (Fig. 4e, f). The number of replicates has a profound impact on the number of significantly regulated sites, which seems to increase linearly as a function of the number of replicates (Fig. 4g). Analyzing four replicates with 90-min runs already results in more significantly regulated sites than three 290-min measurements, while six replicates of 90 min even double the number of significantly regulated sites. For our biological comparison at hand however, we aim to compare the inherent quantification characteristics of the approaches and would further expect similar benefits from more replicates for SILAC and TMT as well. Thus, while we encourage users to measure more than three replicates especially for LFQ, we settled

**Fig. 3** Evaluation of quantification methods in a biological setting. **a** Non- or SILAC-labeled U2OS cells were treated with 5 μM doxorubicin (DOX), 2.5 μM 4-nitroquinoline 1-oxide (4NQO) or DMSO (C) for 2 h before lysis. Three biological replicates were measured for all quantification methods. For MS measurement, each quantification method was given a total of 2 days instrument time (including LC overhead). SILAC samples were fractionated into ten fractions per sample on an Ultimate 3000 high-flow system, and TMT into 24 fractions total on an Ultimate 3000 micro-flow system. Samples were then measured using a 15- or 50-cm (only LFQ) column on a Q Exactive HF or Orbitrap Fusion Lumos (only TMT $MS^3$ OT MC). For SILAC and TMT, MS samples were injected without dilution, so that each labeling channel resembles one LFQ injection. **b** Bar plot showing total numbers of identified and quantified phosphopeptides for all replicates of each quantification method, respectively. Calculations of ratios were performed within biological replicates and filtered for measurement in a minimum of one, two or three replicates, and >75% confident phosphorylation site localization. For further analysis, ratios quantified in all three replicates only and with a localization probability of at least 75% (black arrows) were used. **c** SAM-based identification of significantly regulated phosphorylation sites was performed with two sample paired $t$-test and standard settings (s0 estimation automatic, delta estimation based on FDR = 0.20). Significantly regulated phosphorylation sites (sig) are highlighted in red, non-significant sites in gray. Applied s0 and delta values, as well as the total number of tested phosphorylation sites (*n*) are shown. For LFQ and SILAC nearest neighbor imputation (IMP), phosphorylation sites quantified in at least one replicate and with a localization probability of at least 75% were used. **d, e** The bar plots show the number of significantly regulated phosphorylation sites for each quantification method **d** in total, and **e** as a fraction relative to the total number of tested sites. **f, g** The Venn diagrams show the overlap of SAM-regulated phosphorylation sites identified **f** in total, and **g** for commonly identified sites

to compare SILAC and TMT to the in our hands best-performing LFQ approach with 290 min on a 50-cm column with three biological replicates each.

The analysis of treatment with 4NQO vs. control yielded overall the same conclusions as doxorubicin (Supplementary Fig. 4a−d), but since 4NQO induces a weaker rewiring of the phosphoproteome, we identified less significantly regulated sites for each of the quantification methods compared to doxorubicin. Importantly, the observed loss in both MS[3]-based TMT phosphoproteome coverage and identification of significantly regulated phosphorylation sites compared to MS[2] is not restricted to our data set, but is also observed when reanalyzing published data. Huang et al. published a data set with complementary MS[2]/MS[3]-based TMT quantification of breast cancer cell lines[47].

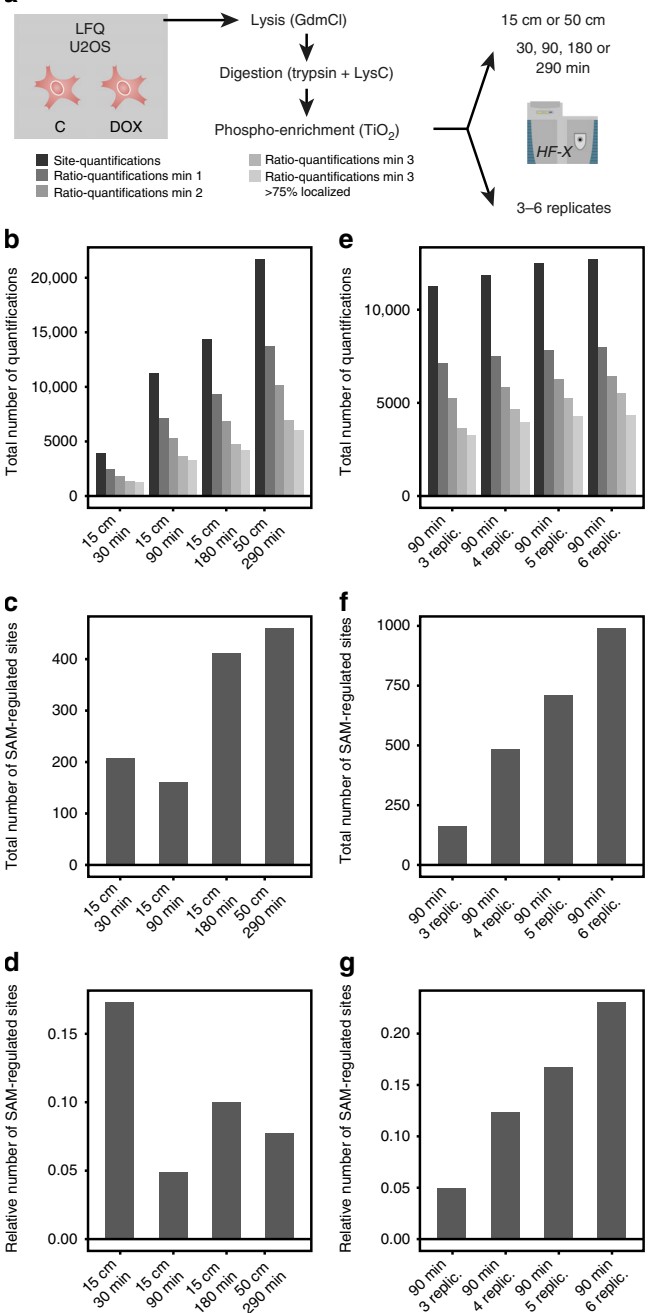

Comparing the basal phosphoproteome of cell lines AU565 vs. T47D in technical duplicates confirms that while MS[3]-based TMT decompresses MS[2]-measured ratios, the latter allows the identification of more significantly regulated phosphorylation sites (Supplementary Fig. 5).

**Quantification methods yield different biological insights**. After concluding that the four quantification methods can identify different numbers of SAM-regulated phosphorylation sites, we wanted to assess if and to which degree these sites gave us biological insight into the cellular signaling of the doxorubicin-induced DDR. Linear sequence motif analysis of the upregulated phosphorylation sites revealed that all techniques, including MS[2]-based TMT, could correctly identify the DDR-induced ATM/ATR kinase substrate motif [s/t]Q as significantly enriched (Fig. 5a)[48]. This is also true when performing linear kinase motif enrichment analysis within Perseus (Fig. 5b). However, when analyzing enriched gene ontology (GO)-terms among the significantly upregulated phosphorylation site ratios, LFQ was not able to identify any DDR-related terms containing the keywords checkpoint, damage, repair, cell cycle or chromosome (Fig. 5c). Only with MBR was LFQ able to identify terms such as "response to DNA damage stimulus" or "recombinational repair", which SILAC could with and without MBR REQ. Neither approach however profited from missing value imputation, which like LFQ alone did not yield any significantly enriched GO terms. Importantly, both TMT methods performed a lot better, yielding a broad variety of DDR-related terms. The deepest coverage of GO terms with most significant q-values was achieved by the MS[2]-based TMT method. The poor performance of LFQ was not simply due to the broader phosphorylation site coverage of TMT. Both TMT-approaches and SILAC still outperformed LFQ when only sites quantified in all eight quantification approaches were used for kinase motif enrichment (Supplementary Fig. 6). Importantly, the 4NQO-based DDR phosphorylation landscape yielded the same conclusions (Supplementary Fig. 4e, f).

**TMT multiplexing enables accurate stoichiometry calculation**. In addition to the identification of significantly regulated sites, calculation of absolute phosphorylation site stoichiometry can give an extra layer of insight into cellular signaling[15–17]. In

**Fig. 4** Identification of significantly regulated phosphorylation sites using LFQ. **a** U2OS cells were treated with 5 μM doxorubicin (DOX) or DMSO (C) for 2 h before lysis. For the gradient experiment (**b**−**d**), samples were measured in three biological replicates using a 15-cm column with a 30, 90 or 180-min gradient, or a 50-cm column with a 290-min gradient on a Q Exactive HF-X. The shorter gradients are all time-compressed versions of the 290-min gradient, and all other LC-MS instrument settings were kept identical between conditions. For the number of replicates experiment (**e**−**g**), samples were measured in six replicates using the 90-min gradient setup on a Q Exactive HF-X, and 3−6 biological replicates (replic.) were used for statistical analysis. **b**, **e** Bar plots showing total numbers of identified and quantified phosphopeptides for the depicted gradients and number of replicates. Calculations of ratios were performed within biological replicates and filtered for measurement in a minimum of one, two or three replicates, and >75% confident phosphorylation site localization. For further analysis, ratios quantified in all three replicates only and with a localization probability of at least 75% were used. **c**, **f** The bar plots show the total number os significantly regulated phosphorylation sites. **d**, **g** The bar plots show the number of significantly regulated phosphorylation sites as a fraction relative to the total number of tested sites. SAM-based identification of significantly regulated phosphorylation sites was performed with two sample paired *t*-test and standard settings (s0 estimation automatic, delta estimation based on FDR = 0.20)

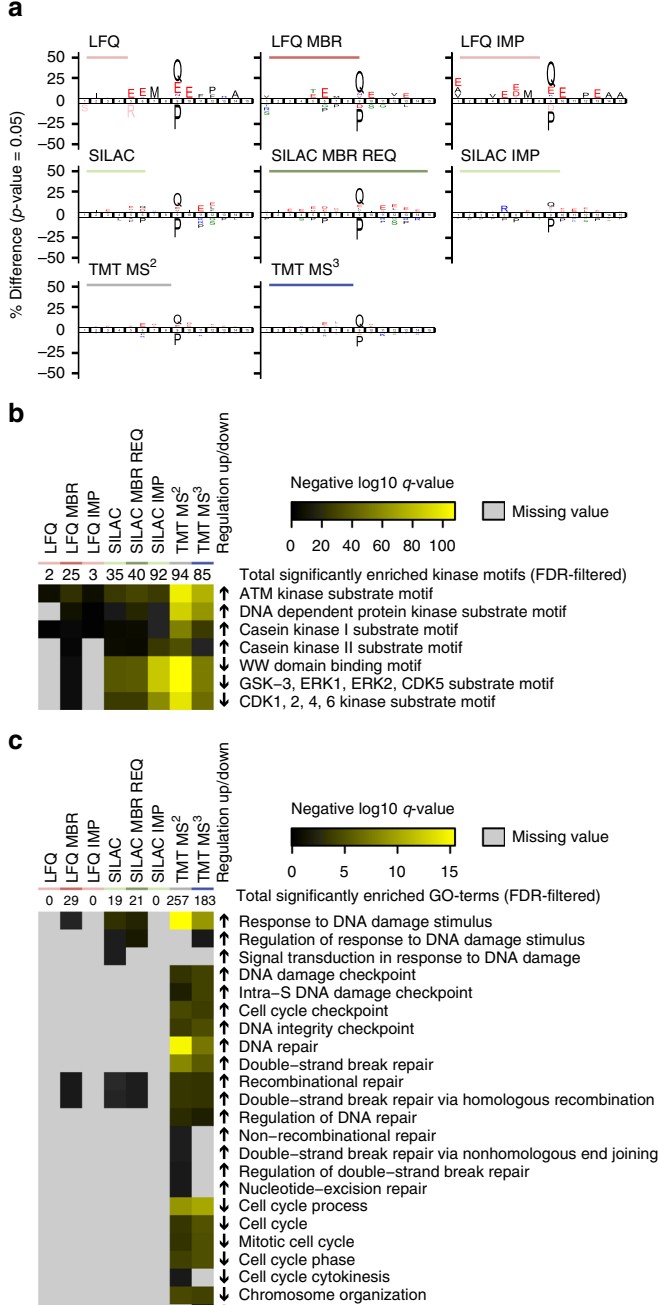

**Fig. 5** Functional characterization of significantly regulated phosphorylation sites. **a** iceLogos of the SAM-upregulated phosphorylation sites from Fig. 3c vs. the respective non-regulated sites as background. The iceLogos show the ATM/ATR kinase substrate [s/t]Q motif significantly enriched for all tested quantification approaches. **b**, **c** Heat maps showing **b** a kinase motif and **c** GO-term enrichment of significantly SAM-up/downregulated phosphorylation sites from Fig. 3c vs. the respective non-regulated sites as background. Enrichment was performed using Fisher exact tests within Perseus with relative enrichment on gene level and an FDR of 0.02. The numbers above the heatmap show the total number of enriched motifs/GO-terms, while the heat maps below show **b** the most significantly regulated motifs or **c** all GO-terms with "damage", "repair", "checkpoint", "cell cycle", or "chromosome", indicative of an activated DDR, respectively

contrast to mere phosphopeptide ratios, stoichiometry can provide information on the change of the phosphorylation status of individual sites relative to the total protein level. Stoichiometry can then be used as an additional significance cutoff, or to differentiate between extreme and minor phosphorylation changes. Phosphorylation stoichiometry can be extracted directly from large-scale quantitative phosphoproteomics experiments by using ratios observed in both the phosphopeptide, its non-phosphorylated counterpart and the respective protein between treatment conditions[15], or directly from the phosphopeptide within a single treatment condition[16,17]. However, we reasoned that the multiplexing capability of TMT should allow the extraction of stoichiometry from multiple treatment conditions at the same time. By integrating information of several treatment conditions and replicates into one stoichiometry model, overall quantification precision should be increased in comparison to calculations based on individual ratios. Thus, instead of equations using ratios, we here developed a 3D multiple regression model (3DMM)-based approach, which uses phosphopeptide-, non-phosphorylated peptide- and corresponding protein-intensities from any multiplexed quantification method, including $MS^2$- and $MS^3$-based TMT experiments (Fig. 6a, Supplementary Note 1 and Supplementary Data 1 and 2).

Since our previous results highlighted the equal importance of quantification accuracy and precision for the identification of significantly regulated sites, we wondered if this also held true for the calculation of phosphorylation site stoichiometry if quantified via TMT. To test this, we prepared a mixed species sample with fixed phosphopeptide stoichiometry (Fig. 6b). After phosphopeptide enrichment from both yeast and HeLa, half of both samples was dephosphorylated using alkaline phosphatase. Mixing together phosphorylated and non-phosphorylated yeast peptides in fixed ratios yielded conditions ranging from 10 to 90% phosphorylation site stoichiometry within a single TMT10-plex sample.

When measuring these samples in both $MS^2$- and $MS^3$-mode, we found that we can assess the quality of the 3DMM linear fit by calculating a $p$-value, which describes the significance of the slope being non-zero. We then show that this $p$-value, which can be calculated for each 3DMM individually, is a reliable determinant of stoichiometry accuracy (Fig. 6c). When comparing $MS^2$- and $MS^3$-based TMT quantification, we can use this $p$-value to filter inaccurate stoichiometry information. This turned out to be important especially for $MS^2$-based TMT measurement. In contrast to identifying significantly regulated phosphorylation sites, quantification accuracy seems to be crucial for accurately estimating phosphorylation site stoichiometry (Fig. 6d). Even though stoichiometry estimated by $MS^2$-based TMT quantification is trending towards the correct value, the estimation accuracy is very low. It can be improved by setting stricter $p$-value cutoffs, but this comes at the expense of excluding an increasing number of identified phosphorylation sites. In contrast, $MS^3$-based TMT quantification-derived stoichiometry is highly accurate even without any $p$-value cutoffs. For example, extreme target occupancies of 10 and 90% were estimated as 12.4% ± 4.0% and 86.7% ± 3.8% (median ± MAD), respectively. Notably for both $MS^2$- and $MS^3$-based TMT quantification, stoichiometry estimation is more accurate and precise at higher stoichiometry values (Fig. 6e).

## Discussion

Quantification methods for proteomics have been evaluated before, but no study compared their application for large-scale phosphoproteomics in a complex biological setup. In this study, we show that the highest accuracy alone does not automatically

guarantee the best performance in cell signaling studies. We found that quantification precision and phosphoproteome coverage can be equally important. That is why, even with high ratio compression, MS²-based TMT quantification was able to identify more than twice as many significantly regulated phosphorylation sites than MS³-based TMT methods based on multiple testing-corrected SAM-testing. Of course, more significant hits do not imply better quantification by themselves. However by

demonstrating their meaningful representation of the expected DDR, we argue that they are a good indicator of the quantification performance. Our data also shows that this increase in significant hits is caused by the higher phosphopeptide coverage of MS²-based TMT, facilitated by its faster peptide scanning speed and its higher apparent precision. The higher apparent precision seems to indeed enable robust peptide quantification for MS²-based TMT, as demonstrated by its good compromise of TPR vs.

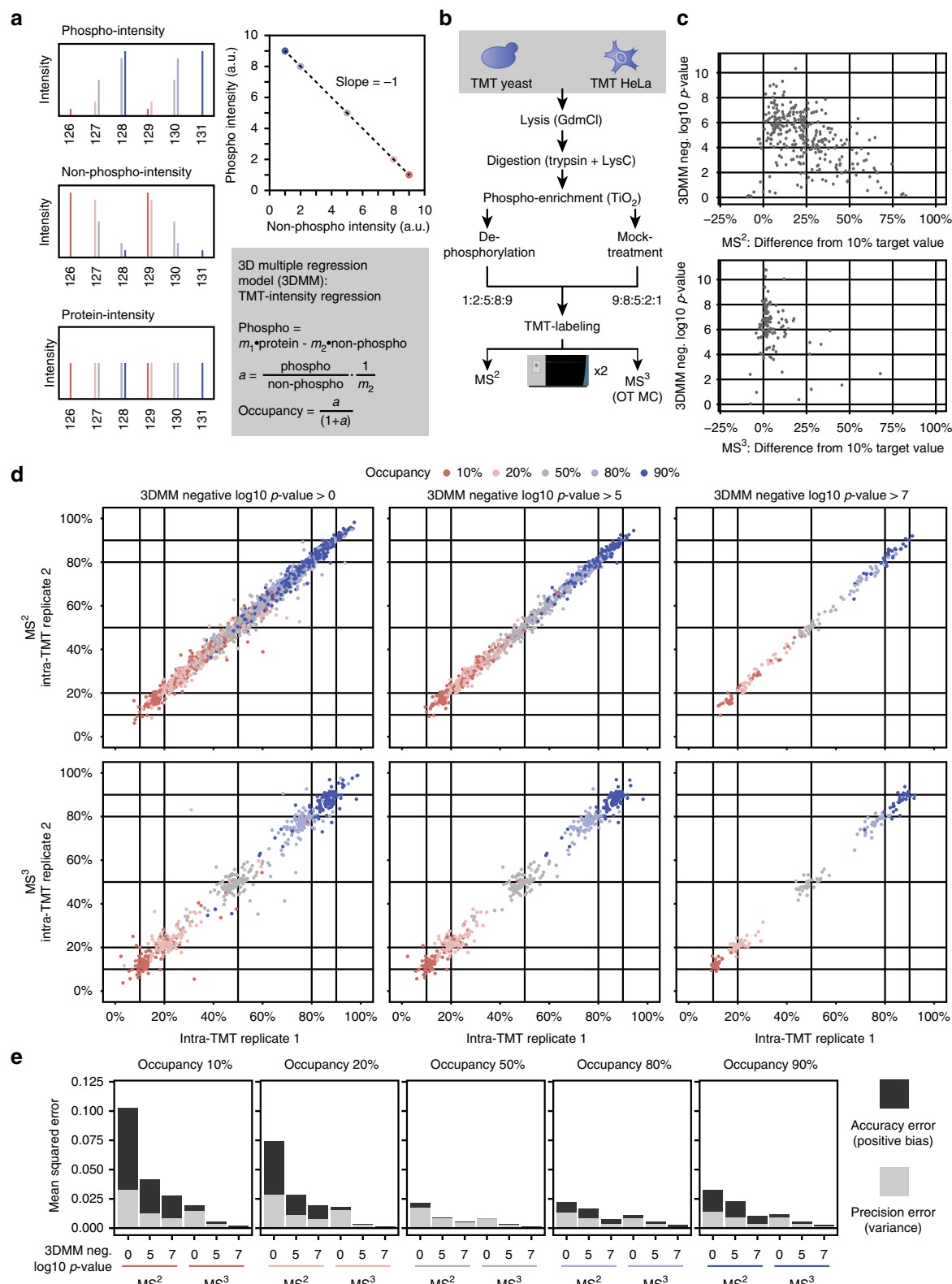

FPR in our ROC curve analysis. Interestingly, we found that the higher accuracy of SPS-MS$^3$-based TMT approaches is directly negated by its lowered precision. We assume that this decrease is caused by the generally lower signal-to-noise of TMT reporter ions on MS$^3$ level. For complex cell signaling studies, where a minimum of three biological conditions and three biological replicates, as well as offline peptide fractionation are currently standard requirements, MS$^2$-based TMT quantification furthermore outperforms SILAC. Even though the latter shows the best compromise between accuracy and precision, it suffers from lower phosphopeptide coverage and low multiplexing capabilities. Our data furthermore indicates that LFQ is the least suitable quantification method for cell signaling studies among the ones we tested, due to its lower precision and missing multiplexing capabilities. However, this disadvantage might be counterbalanced by activating MBR or measuring more replicates instead.

Apart from comparing quantification approaches of significantly regulated phosphorylation sites, we also present a quantitative benchmark setup for global analysis of phosphorylation site stoichiometry. We applied it by adapting stoichiometry estimation to TMT-based quantification, using a newly developed 3D multiple regression model-based approach, which takes advantage of the high multiplexing capabilities of TMT. A direct comparison of MS$^2$- and MS$^3$-based analysis highlighted that in this context, the high accuracy of MS$^3$-based TMT quantification is crucial for achieving accurate and reliable stoichiometry information. In addition, we found that $p$-values extracted from the fit of our 3DMM can estimate the quality and serve as cutoff values, which is especially useful for MS$^2$-based TMT quantification. It has to be noted that our benchmark setup made it necessary to simulate unregulated protein intensities for the 3DMM. Furthermore, the approach most likely shares the disadvantage of SILAC stoichiometry estimation[15], which is that subtle ratio changes are difficult to detect. However, even though we did not test this with our setup, MS$^3$-based TMT multiplexing over 11 treatment conditions, instead of three in SILAC, makes it more likely that a biological change is observed, and further increases precision of the modeling. We thus believe it highlights the potential of linear modeling-based TMT data analysis, not only for phosphorylation stoichiometry, but for other PTM applications as well.

For the purpose of this comparative analysis, we believe that the different combinations of offline fractionation and LC-MS/MS settings we eventually used represent our best optimized setups for each respective method. We are of course aware that alternative fractionation strategies, MS settings or biological systems can yield slightly different results. Nevertheless, we argue that our conclusions hold true, not least because we could confirm our results for MS$^2$- and MS$^3$-based TMT by reanalyzing a published data set from Huang et al.[47]. Furthermore, as multiplexing is not possible for LFQ and routinely only up to three channels for SILAC, their accuracy, precision or identification rates would need to increase substantially to catch up to a TMT-

based workflow with offline peptide fractionation. Even when analyzing more than 11 conditions, where current-generation TMT would start suffering from missing values as well, this issue would be even more severe for LFQ and SILAC. With recent developments in LFQ-based data independent acquisition (DIA) quantification[46], it might be interesting to see how this approach can compare to TMT multiplexing for quantitative phosphoproteomics experiments, once current DIA limitations such as reliable phosphorylation site localization can be routinely addressed. To increase phosphopeptide coverage and quantification precision of MS$^3$-based over MS$^2$-based TMT, one would need to increase scanning speed and/or identification rates, and signal-to-noise ratios on MS$^3$ levels, respectively. There is currently no evidence that a new MS$^3$-based method that overcomes these obstacles will be routinely available in the foreseeable future. Instead, future developments in alternative quantification approaches such as complementary reporter ion readout, or gas-phase- or ion mobility-based ion purification may enable precise and accurate large-scale phosphopeptide quantification[34,35,37]. In the meantime, we advise large-scale phosphoproteomics users to consider using MS$^2$-based TMT-based quantification, as long as reproducible but not necessarily accurate quantification is required.

## Methods

**Human cell culture.** All experiments were performed as either technical (Figs. 1, 2 and 6 and Supplementary Fig. 2) or biological (Figs. 3 and 4) replicates. Human epithelial cervix carcinoma HeLa cells (female) and human epithelial osteosarcoma U2OS cells (female) were purchased from ATCC. Cells were cultured in DMEM high glucose with Glutamax (Gibco, 31966–021) or for SILAC experiments in DMEM high glucose without L-glutamine, lysine and arginine (Biowest, A0480–500), both with 10% fetal bovine serum (Gibco, 10270–106) and 100 U/ml penicillin/streptomycin (Invitrogen, 15140-122) at 37 °C in a humidified incubator with 5% CO$_2$. For SILAC experiments with and without yeast protein background, cells were labeled with three different isotopic versions of lysine ("0": normal Lys, "4": Lys-D$_4$, "8": Lys-$^{13}$C$_6$,$^{15}$N$_2$), or lysine and arginine ("0": normal Arg, "6": Arg-$^{13}$C$_6$, "10": Arg-$^{13}$C$_6$,$^{15}$N$_4$), respectively (Cambridge Isotope Laboratories Inc., CNLM-291-H-PK)[21]. We have not performed specific authentication of the cell lines in this study. Cells were tested mycoplasma negative via PCR-testing. Treatment with genotoxic agents doxorubicin (Sigma Aldrich, D1515-10MG) and 4-nitroquinoline 1-oxide (Sigma Aldrich, N8141) with DMSO (Sigma-Aldrich, D8418–250ml) as a control were performed for 2 h at final concentrations of 5 μM and 2.5 μM diluted in DMSO, respectively. Cells were harvested with or without previous treatment at approximately 90% confluency by washing twice with PBS (Gibco, 20012–068) and then adding 95 °C hot GdmCl lysis buffer (6 M guanidine hydrochloride, Sigma-Aldrich, G3272–2KG; 5 mM tris(2-carboxyethyl)phosphine, Sigma-Aldrich, C4706–10G; 10 mM chloroacetamide; 100 mM Tris, pH 8.5, Sigma-Aldrich, 10708976001) supplemented with protease and phosphatase inhibitors (1 complete mini protease inhibitor cocktail tablet, Roche, 04693124001; 50 mM sodium fluoride; 10 mM sodium orthovanadate; 50 mM β-glycerol phosphate, Sigma-Aldrich, G5422). After rocking for 5 min, cells were scraped and lysate was boiled for 10 min at 95 °C. DNA was sheared by 2-min ultrasonication treatment (Sonics & Materials, VCX 130; 1 s on, 1 s off, 80% amplitude).

**Yeast cell culture.** BY4742 wt yeast cells were grown in yeast medium (drop out mix without lysine, Nordic Biosite, D9515B; 6.7 g/l yeast synthetic drop-out medium supplements without lysine, Sigma-Aldrich, Y1376–20G; 2% v/v glucose, Sigma-Aldrich, G7021–1KG; 205 µl/l SILAC lysine 0/4/8) at 30 °C and 200 rpm rotation in overnight cultures. Day cultures were inoculated at OD$_{600}$ of ~0.1 and harvested at OD$_{600}$ of ~0.9. Yeast cells were washed with ice-cold PBS and 1 l of

**Fig. 6** 3D multiple regression model-based calculation of phosphorylation stoichiometry. **a** Phosphorylation stoichiometry can be extracted by feeding phospho-, non-phospho- and protein-intensity data into a 3D multiple regression model (3DMM). More detailed explanations are given in Supplementary Note 1. **b** For benchmarking stoichiometry calculation via MS$^2$- and MS$^3$-based TMT, yeast and HeLa phosphopeptides were each half dephosphorylated with Rapid alkaline phosphatase. Yeast phospho- and non-phospho-peptides were then diluted in fixed ratios to create samples with set phosphopeptide stoichiometry, and added to equal amounts of HeLa phospho- and non-phospho-peptides serving as a contaminating background. The sample was measured three times as technical replicates each with MS$^2$- and OT MC MS$^3$-based TMT quantification. In this setup, protein intensities were set to 1 in the 3DMM. **c** 3DMM-extracted $p$-values describing the significance of the slope being non-zero were correlated against the difference of MS$^2$- and MS$^3$-estimated stoichiometry vs. the true value of 10%. **d** Scatter plots showing estimated stoichiometry determined in TMT MS$^2$ and MS$^3$ mode, with three different levels of 3DMM $p$-value cutoffs. **e** Mean squared errors were calculated as a sum of positive bias and variance for all replicates of both MS$^2$- and MS$^3$-based TMT at different 3DMM $p$-value cutoffs

OD$_{600}$ 0.5 cells equivalent were resuspended in 10 ml yeast lysis buffer (75 mM TrisCl pH 8.0; 75 mM NaCl, Sigma-Aldrich, S5886–1KG; 1 mM ethylenediamine-tetraacetic acid, Sigma-Aldrich, ED-500G; protease and phosphatase inhibitors listed above). Resuspended yeast was frozen in droplets in liquid nitrogen, and grinded in a MM400 ball mill (Retsch) for 3 min at 25 Hz. Yeast pellet powder was supplemented with 1% Triton X-100 (Sigma-Aldrich, T9284) v/v and thawed at 4 °C for 30-min rolling. Debris was spun down, and supernatant was transferred to −80 °C acetone (Merck, 1.00020.2500) to a final 80% v/v and incubated for 4 h at −20 °C. Precipitated proteins were spun down and resuspended in 95 °C GdmCl lysis buffer and treated as human cells written above.

**Protein digestion.** Protein concentration was estimated by BCA assay (Thermo Fisher Scientific, 23225). 1–2 mg protein per condition was digested with 1:100 w/w LysC (Wako, 129-02541) for 4 h at 37 °C. Samples were diluted 1:3 in 25 mM TrisCl pH 8.5 and digested with 1:100 w/w trypsin (Sigma-Aldrich, T6567) at 37 °C overnight. After acidification 1:10 with 10% TFA (Sigma-Aldrich, T6508-500ML) and spinning down 2 min at 2000 × g, peptides were purified on SepPak Classic C18 cartridges (Waters; WAT051910).

**High flow fractionation for SILAC biological benchmark only.** Prior to phosphopeptide enrichment, SILAC samples were fractionated according to a modified protocol by Batth et al.[11]. We used a Waters XBridge BEH130 C18 3.5 µm 4.6 × 250 mm column on an Ultimate 3000 high-pressure liquid chromatography (HPLC) system (Dionex) operating at a flow rate of 1 ml/min with three buffer lines: Buffer A consisting of water, buffer B of ACN (Merck, 1.00030.2500) and buffer C of 25 mM ammonium bicarbonate, pH8 (Sigma-Aldrich, 09830-500G). Peptides were separated by a linear gradient from 5% B to 25% B in 50 min followed by a linear increase to 75% B in 5 min, and kept there for 5 min before ramping to 5% in 5 min. Buffer C was constantly added to the gradient at 10%. Ten concatenated fractions consisting of pooled 1-min fractions were collected and then used for phosphopeptide enrichment.

**Phosphopeptide enrichment.** Phosphopeptides were enriched from samples using titanium dioxide beads (TiO$_2$; GL Sciences, 5020-75000) according to a modified protocol from Jersie-Christensen et al.[49]. TiO$_2$ beads were pre-incubated in 2,5-dihydroxybenzoic acid (20 mg/ml; Sigma-Aldrich, 85707-1G-F) in 80% ACN/1% TFA (5 ml/mg of beads) for 20 min. All fractions were brought to 80% ACN and 5% TFA in a final volume of 5 ml. Beads equivalent to 2× starting protein amount (in 5 ml of DHB solution) were added to each sample, which were then incubated for 20 min while rotating. Beads were transferred to C8 StageTips (made from Sigma Aldrich, 66882-U)[50] and washed with 10% ACN/6% TFA, 40% ACN/6% TFA, and 60%/6% TFA. Phosphopeptides are then eluted with 5% ammonia (Merck, 1054321011) and 10% ammonia/25% ACN and subsequently loaded onto C18 StageTips (made from Sigma-Aldrich, 66883-U). Peptides were eluted with 40 and 60% ACN and subjected to TMT labeling or directly to MS measurement.

**TMT labeling.** Enriched phosphopeptides were mixed with HEPES at pH 8.5 (Sigma-Aldrich, H3375) to a final concentration of 50 mM, as suggested by Ting et al.[32]. TMT10-plex reagents (Thermo Fisher Scientific, 90110) were solubilized in acetonitrile according to the manufacturer's instructions. We performed dilution experiments and determined that 1 µl TMT reagent is sufficient to label phosphopeptides equivalent to ~1 mg protein starting material to reach a labeling efficiency of >95% for the cell lines used in this publication. We confirmed for randomly selected raw files from all five TMT data sets in this study that the labeling efficiency was >97%. After vortexing and incubating for 1 h at room temperature, reactions were quenched using a 5% hydroxylamine solution (Sigma-Aldrich, 467804-10ML) at 1 µl per 8 µl TMT reagent. After further 15-min incubation, the peptide solutions were acidified 1:10 v/v with 10% formic acid (Merck, 1.00264.1000) and loaded onto C18 StageTips. Peptides were either eluted with 40 and 60% ACN for MS measurement, or with 40 and 60% ACN in 25 mM ammonium bicarbonate for micro-flow fractionation.

**Micro flow fractionation for TMT biological benchmark only.** Phosphopeptides were fractionated using a Waters Acquity CSH C18 1.7 µm 1 × 150 mm column on an Ultimate 3000 HPLC system (Dionex) operating at a flow rate of 30 µl/min with two buffer lines: Buffer A consisting of 5 mM ammonium bicarbonate and buffer B of 100% ACN. Peptides were separated by a linear gradient from 5% B to 25% B in 62.5 min followed by a linear increase to 60% B in 4.5 min and 70% B in 3 min, and kept there for 7 min before ramping to 5% in 1 min. Twenty-four concatenated fractions consisting of pooled fractions of variable time length were collected and directly subjected to MS measurement. For MS$^3$-based TMT, concatenation of two fractions failed, which thus had to be measured individually, resulting in a total of 26 equivalent measured fractions.

**Nanoflow LC tandem MS.** All samples were analyzed on an Easy-nLC 1000 coupled to a Q-Exactive HF instrument (Thermo Fisher Scientific; TMT MS$^2$ of Fig. 3), an Orbitrap Fusion Lumos instrument (Thermo Fisher Scientific; Figs. 1, 2, TMT MS$^3$ of 3, 6, Supplementary Fig. 2), or a Q-Exactive HF-X instrument

(Thermo Fisher Scientific; Fig. 4), all equipped with a nanoelectrospray source. Peptides were separated on a 15-cm or 50-cm (Fig. 2 and LFQ Fig. 3 only) analytical column (75-µm inner diameter) in-house packed with 1.9-µm C18 beads (Dr. Maisch, r119.b9). The column temperature was maintained at 40 °C or 50 °C for the 15-cm and 50-cm column, respectively, using an integrated column oven (PRSO-V1, Sonation). For SILAC and TMT in the technical comparison in Fig. 2, MS samples were diluted to contain a total peptide amount equal to one LFQ injection based on protein starting amount, to enable MS intensity-independent comparison of the method-inherent quantification characteristics. For the biological comparison in Fig. 3 and the original technical comparison on Supplementary Fig. 2c–e, SILAC and TMT MS samples were injected without dilution, so that each labeling channel resembles one LFQ injection. Each peptide fraction was auto-sampled and separated using gradients optimized for the type of sample, the column length and the available MS time. We found that TMT-labeling seemed to make the peptides more hydrophobic and tried to optimize our gradients based on this observation. For the TMT MS$^2$/MS$^3$ comparison (Fig. 1), MS$^2$- and MS$^3$-based TMT of the DDR comparison (Fig. 3), and the occupancy comparison (Fig. 6), we used a 90-min gradient at a flow rate of 250 nl/min ramping from 10% buffer B (80% ACN and 0.1% formic acid) to 30% B in 69 min, to 45% B in 13 min, to 80% B in 2 min, kept 2 min, to 5% B in 2 min and kept 2 min. For the accuracy comparison (Fig. 2 and Supplementary Fig. 2) and LFQ of the DDR comparison (Fig. 3), we used a 290-min gradient ramping from 5% B to 30% B in 240 min, to 80% B in 35 min, kept 5 min, to 5% B in 5 min and kept 5 min. The shorter LFQ gradients for 30, 90, and 180 min on a 15-cm column in Fig. 4 are time-compressed versions of this 290-min gradient. For SILAC of the DDR comparison (Fig. 3), we used a 70-min gradient ramping from 10% B to 30% B in 54 min, to 45% B in 10 min, to 80% B in 1 min, kept 2 min, to 5% B in 1 min and kept 2 min. The mass spectrometers were operated in DDA mode to automatically isolate and fragment topN multiply charged precursors according to their intensities. Detailed MS settings for the methods used in this study are listed in Supplementary Table 1.

**Raw data processing.** An overview of all raw LC-MS/MS files is given in Supplementary Data 3. All raw LC-MS/MS data were processed with MaxQuant[13] v1.5.5.4i and v1.5.8.0 (only for Supplementary Fig. 3) using the Andromeda search engine and searched against the complete human UniProt database including all Swiss-Prot entries (downloaded 2016-04-14), and in case of mixed human/yeast samples additionally with the complete yeast UniProt database including all Swiss-Prot entries (downloaded 2016-11-30). In addition, the default contaminant protein database was included. The "match between runs" (MBR) and SILAC requantify (REQ) features were activated where indicated. As activating MBR for SILAC showed essentially no differences to having it deactivated, we did not show this data in this study. Data sets for LFQ, SILAC, MS$^2$- and MS$^3$-based TMT, as well as LFQ-MBR and SILAC-MBR REQ were kept in individual MaxQuant analysis groups (Supplementary Data 3). TMT correction factors in MaxQuant were updated to the values provided by the manufacturer. For LFQ of the DDR comparison (Fig. 3), four replicates were measured, but only three were used at random in the data analysis, equal to SILAC and TMT. Carbamidomethylation of cysteine was specified as fixed modification for all groups. Variable modifications considered were oxidation of methionine, protein N-terminal acetylation, and phosphorylation of serine, threonine and tyrosine residues.

**False discovery rate analysis.** False discovery rate (FDR) filtering was applied as described before[12,13]. Briefly, the FDR was set to 1% on peptide spectrum match (PSM), PTM site and Protein level. MaxQuant applies a target-decoy search strategy to estimate and control the extent of false-positive identifications using the concept of posterior error probability (PEP) to integrate multiple peptide properties, such as length, charge, number of modifications, and Andromeda score into a single quantity reflecting the quality of a PSM.

**Bioinformatics analysis.** The majority of the data analysis was accomplished by using custom R scripts with R 64 bit version 3.4.0[51], including the R package data table v1.10.4[52]. Signal-to-noise ratios were extracted from MS raw files using raxport.exe v3.3 (http://www.findbestopensource.com/product/raxport)[53] and a custom Perl script using ActivePerl v5.24.0.2400, with the script being provided as Supplementary Data 4. All intensities, except for the technical benchmarks (Fig. 2 and Supplementary Fig. 2), were quantile normalized using the R package preprocessCore v1.40.0[54]. For the technical benchmarks, this would have affected the quantification accuracy estimation, which is why intensities were not normalized here, except for SILAC total triplet intensities. Furthermore for the technical benchmarks, SILAC incorporation efficiency was corrected by using missed cleavage-separated correlation factors between MaxQuant raw and "normalized" SILAC ratio columns, extracted from a 1:1:1 mixture of HeLa or yeast, respectively. SILAC intensities were calculated from total SILAC intensities and SILAC ratios, as these proved to be slightly more robust in SAM analysis than the default MaxQuant-calculated SILAC intensities. For phosphorylation site analysis, the lowest available underscore intensity entries from the MaxQuant output were used. For phosphorylation localization, the lowest MaxQuant-calculated localization probability per method was used to filter confidently localized phosphorylation

sites with a threshold of >0.75. However, MBR-derived localization scores of 0 were ignored.

Heat maps were created using the R package gplots v3.0.1[55]. Bar plots, box plots, scatter plots, violin plots and ROC plots were creating using the R package ggplot2 v2.2.1[56]. Venn diagrams were created using the R package VennDiagram v1.6.17[57]. Phosphorylation site stoichiometry information was calculated using a custom R script, which is appended as Supplementary Data 1, together with example data from Fig. 6 in Supplementary Data 2. The theoretical background of multiplexed stoichiometry calculation is described in more detail in Supplementary Note 1. After calculation of raw stoichiometry, "illegal stoichiometry", i.e. values $x$ outside the boundary of $0 <= x <= 1$, were excluded from further analysis.

**Statistics.** SAM analysis was performed either in R with the R package SAMR v2.0[45] (two class paired or unpaired $t$-testing, automatic s0 and FDR-based delta-determination, a random seed = 123, and default values or otherwise indicated for the FDR), or in Perseus v1.6.0.7[14] (two-sided $t$-test with FDR-adjustment of a q-value threshold of 0.05, and s0 set to 0.1 or 0.2). Supplementary Data 5 contains all normalized intensities from the biological benchmark and the results from SAM testing (Fig. 3). Where indicated, imputation was performed with the R package "impute" incorporated in the "samr" package using standard settings. Sequence motif logos were generated using iceLogo v1.2[58] with fold-change as the scoring system and a $p$-value cut-off of 0.05. Our input data sets were sequence windows for SAM upregulated phosphorylation sites of each quantification method. For the background data sets, we used all phosphorylation site ratios used for SAM testing for each method, respectively. Significant enrichment of kinase linear motifs or GO terms was performed using the Fisher exact test within Perseus, with standard settings of FDR = 0.02 and relative enrichment on gene name level. Supplementary Data 6 contains all enriched motifs and GO-terms from the biological benchmark based on the results from SAM testing (Fig. 5).

**Code availability.** Custom R code to perform 3DMM stoichiometry calculations in Fig. 6 is available as an R script in the supplementary section as Supplementary Data 1 and example data from Fig. 6 as Supplementary Data 2. All other custom R codes are available upon request.

**Data availability.** All raw MS data files from this study have been deposited to the ProteomXchange Consortium[59] via the PRIDE partner repository[60] with identifier PXD007145 (https://www.ebi.ac.uk/pride/archive/projects/PXD007145). Raw MS data files from Huang et al. (2017)[47] were downloaded from MassIVE with identifier MSV000079655 (ftp://massive.ucsd.edu/MSV000079655). All other data supporting the findings of this study are available from the corresponding author on reasonable request.

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

## Acknowledgements

We would like to acknowledge Rosa Viner (Thermo Fisher Scientific) for providing Orbitrap Fusion Lumos acquisition methods. We would like to thank Chiara Francavilla, Rosa Jersie-Christensen and Tanveer Batth for critical input on the project. Work at The Novo Nordisk Foundation Center for Protein Research (CPR) is funded in part by a generous donation from the Novo Nordisk Foundation (grant number NNF14CC0001). The proteomics technology developments applied was part of a project that has received funding from the European Union's Horizon 2020 research and innovation programme under grant agreement no. 686547. We would like to thank the PRO-MS Danish National Mass Spectrometry Platform for Functional Proteomics and the CPR Mass Spectrometry Platform for instrument support and assistance. J.V.O. Group was supported by the Danish Cancer Society (R90-A5844 KBVU project grant) and Lundbeckfonden (R191-2015-703).

## Author contributions

A.H. designed and performed all experiments, and analyzed the data and wrote the paper. L.v.S. contributed to the experimental design on the D.D.R. experiments and provided input on the manuscript. D.B.B.-J. helped set up the microflow high pH fractionation system. B.T.W. helped with yeast experiments. C.D.K. helped with data analysis and input on M.S. setup. J.V.O supervised the project, input on project design, data analysis and wrote the paper.

## Additional information

**Competing interests:** The authors declare no competing interests.

