## [Peer Review File · Nature Communications]

Reviewers' comments:

Reviewer #1 (Remarks to the Author):

The study Benchmarking LFQ, SILAC and MS2/MS3-based TMT quantification strategies for large-scale phosphoproteomics sets out to essentially find the best way to perform quantitative phosphoproteomics experiments as well as generating a new occupancy algorithm. Proteomics researchers can now routinely produce high quality data for, often, nearly the entire proteome. Yet, due to poor overlap between runs and reduced precision, quantitative phosphoproteomics remains more challenging. The work presented here is timely since this is likely the next area that must excel for broader proteomics applicability. The comparisons included in this study are thorough and well designed. I think these types of comparisons are exceptionally difficult to perform because each of these quantitative methods is a skill and constantly changing. Comparing optimized with optimized is the best that can be done but I think the results can be often overstated as what works well for one lab may not for another. That said I think it is an important study and may lead to the field examining phosphoproteomics methods more. The study will certainly be useful as a reference.

I didn't find anything particularly surprising except for how poorly LFQ performed. It's not clear to me how missing values were treated. If a phosphorylation exists in only one of the conditions then the site should really only be measurable in that condition. If you throw these out, then you are throwing out the biggest changers. Also I would have suggested doing more replicates with LFQ with shorter run times. We don't see much improvement after 3 hrs, more replicates can somewhat make up for the missing values. I am surprised that more phospho identifications were found with TMT data than the LFQ, even considering the TMT data was fractionated. This doesn't agree with my experience.

I appreciate the authors detailing the different TMT methods they used and what the results were for each. I think this is a big point of confusion for phosphoproteomic work.

The authors state that the precision of TMT is better when using MS2 while noting that this improvement actually comes from precursor interference. This is a common thought that I fundamentally disagree with. The measured precision is assigned to the identified peptide but it's actually a measurement of all the precursors that were co-isolated. The perceived improvement is an artifact and is not necessarily representative of the identified peptide.

The phosphorylation model sample is very clever.

Reviewer #2 (Remarks to the Author):

In their work, Hoglebe et al. compared some of the most popular quantitative MS approaches and discussed their impact on the analysis of the phosphoproteome. As pointed out by the authors, similar comparisons have been done for protein levels; however phosphorylation data might behave differently, given the fact that most of the sites can only be determined on the basis of one peptide measurement. Nevertheless, one of the key points of the analysis (the comparison of MS2 vs MS3 modes in isobaric labeling of phosphopeptides) has been studied recently by others (Erickson et al. and Huang et al) which reduces to some extent the novelty of the work. In addition, the authors also proposed a new strategy to determine the stoichiometry of phosphorylation using the high accuracy achieved with MS3, which is quite significant. The manuscript is well written, the experiments are well executed and the data is presented

transparently. However, the manuscript deals with a very specific technical issue questioning whether the broad audience of Nat Communications is the right target. Alternatively, the manuscript might better fit on a specialized journal wherein will be well received by the MS community, given the technical importance of the question under study here.

Some comments to the authors:

- The authors applied the Significance A to be able to analyze the SILAC data. However, in my opinion, the use of this test in this scenario is incorrect for two reasons. i) the biological triplicates are collapsed into a single ratio, thus the reproducibility of the independent measurements is not considered, ii) this is an outlier test and therefore is influenced by the whole distribution of ratios: it fails identifying significant changes when the two samples under comparison are very different. The latter is observed in the re-analysis of Huang et al. data (Suppl. Fig 5) in which two breast cancer cell lines were compared. While the t-test found 40% of the phosphosites as statistically different, the Significance A only found 1.8%. As an alternative to Sig A, the authors might consider the use of one-sample t-test as a mean to compare all four strategies under the same statistical criteria.
- Figure 4. Why the authors used only the top 1% of Sig A up-regulated sites? Do the conclusions change if the two-sample t-test (at least for LFQ, MS2 and MS3 data) up-regulated sites are used instead?
- Figure 4b. The p-values should be adjusted to q-values (testing the entire GO database can result in a large fraction of false positive enriched terms)
- Line 181: the low precision found on the TMT MS2 data (Fig 2b and 2c) is just a consequence of the nature of the isobaric interference issue, thus it is not strictly correct to state that this strategy yields the highest precision. The variance of these measurements will be inevitably smaller given the significant contribution of the interference (whose variance tends to 0)
- Figure 3c. For the same reason as above, the small variance of the MS2 data favors the identification of statistically significant changes in phosphosites whose mean difference is marginal (and therefore biologically irrelevant). The authors should apply, on top of the q-value, a second criterion based on the magnitude of change, i.e. fold-change (see for instance Gai Gianetto et al. Proteomics 2016)
- Line 210: the authors used a long gradient (5h) for LFQ (to compensate the lack of pre-fractionation in this technique). However, the performance of the chromatography (e.g. peak shapes, peak intensities) may have an impact on MS1-based quantification. Albeit this will reduce the number of identifications, would shorter gradients result in MS1 data of a higher quality and therefore improve the statistical outcome of this approach?

Point-by-point responses to reviewer's comments

Manuscript number: **NCOMMS-17-21440-T**

Our responses to reviewer comments are provided below in **blue font** after each comment and changes to the manuscript are visualized by *italic blue font*.

Dear Reviewers,

We are deeply appreciative for your helpful comments. As you will read, your suggestions have led to a substantial revision including two additional large-scale phosphoproteomics experiments that significantly strengthened the manuscript. We were encouraged to learn that Reviewer #1 is enthusiastic about this work, as he or she found our manuscript dealing with an important issue at the right time and highlights its usefulness as a reference in Nature Communications, and that Reviewer #2 especially appreciates the benchmark of our newly presented phospho-stoichiometry approach.

We have spent the last 2.5 months generating new data, redesigning our analysis and making revisions that endeavor to further improve the quality of the work and address your concerns. Specifically, we are now using the well-established SAM testing, as suggested by reviewer #2, to assess significant regulation for all tested quantification approaches. We use SAM-testing to assess the impact of the higher apparent MS²-based TMT precision in a newly introduced ROC curve analysis, which gives an interesting insight into the specificity of all tested approaches. We further measured a new LFQ data set, to illustrate the role of gradient lengths and number of replicates for the identification of significantly regulated sites.

We discuss how we have attended to your comments point by point, below (marked in blue text).

Overview reviewer's comments

Reviewer #1 (Remarks to the Author):

The study Benchmarking LFQ, SILAC and MS2/MS3-based TMT quantification strategies for large-scale phosphoproteomics sets out to essentially find the best way to perform quantitative phosphoproteomics experiments as well as generating a new occupancy algorithm. Proteomics researchers can now routinely produce high quality data for, often, nearly the entire proteome. Yet, due to poor overlap between runs and reduced precision, quantitative phosphoproteomics remains more challenging. The work presented here is timely since this is likely the next area that must excel for broader proteomics applicability. The comparisons included in this study are thorough and well designed. I think these types of comparisons are exceptionally difficult to perform because each of these quantitative methods is a skill and constantly changing. Comparing optimized with optimized is the best that can be done but I think the results can be often over stated as what works well for one lab may not for another. That said I think it is an important study and may lead to the field examining phosphoproteomics methods more. The study will certainly be useful as a reference.

ANSWER: We thank the reviewer for his or her thorough assessment of our manuscript and for highlighting that it is an important and timely study, which will be useful as a reference for the proteomics community.

I didn't find anything particularly surprising except for how poorly LFQ performed. It's not clear to me how missing values were treated. If a phosphorylation exists in only one of the conditions then the site should really only be measurable in that condition. If you throw these out, then you are throwing out the biggest changers.

ANSWER: We agree that it is important to describe in detail how missing values are treated in our analysis as this can significantly affect the outcome of the data analysis. To address this, we now describe and discuss in detail the issue of 'missing values' and how we deal with it. Please see a detailed explanation of the new results in the paragraph below named **Point 2: LFQ performance and missing values.**

Also I would have suggested doing more replicates with LFQ with shorter run times. We don't see much improvement after 3 hrs, more replicates can somewhat make up for the missing values. I am surprised that more phospho identifications were found with TMT data then the LFQ, even considering the TMT data was fractionated. This doesn't agree with my experience.

ANSWER: We fully agree with the reviewer on this point and thank him or her for the suggestion of performing more replicates of the LFQ experiments with different LC gradients. Reviewer #2 also pointed out the importance of addressing the influence of LC gradient length on the number of significantly regulated sites. We have therefore performed new

label-free phosphoproteomics experiments accordingly. Please see a detailed explanation of the new results in the paragraph below named **Point 3: LFQ gradient length and number of replicates**.

I appreciate the authors detailing the different TMT methods they used and what the results were for each. I think this is a big point of confusion for phosphoproteomic work.

ANSWER: We thank the reviewer for pointing this out. We agree that it is important to specify the details of the different TMT acquisition methods such that other proteomics researchers can reproduce our data by setting up the same LC-MS/MS acquisition methods. We therefore created a detailed table (Supplementary Table 1), which details all important instrument parameters and settings for the six different acquisition methods that we used.

The authors state that the precision of TMT is better when using MS² while noting that this improvement actually comes from precursor interference. This is a common thought that I fundamentally disagree with. The measured precision is assigned to the identified peptide but it's actually a measurement of all the precursors that were co-isolated. The perceived improvement is an artifact and is not necessarily representative of the identified peptide.

ANSWER: We agree that the apparent higher precision of TMT in MS²-based quantification is noteworthy, but needs to be explored in more detail. In an attempt to do this we now performed a number of new experiments and statistical analyses, which are described in detail in the paragraph below named **Point 4: TMT precision**.

The phosphorylation model sample is very clever.

ANSWER: We thank the reviewer for pointing this out. We did attempt to create the ideal phosphorylation model sample for the benchmarking, and are happy to read that the reviewer appreciates this effort.

Reviewer #2 (Remarks to the Author):

In their work, Hoglebe et al. compared some of the most popular quantitative MS approaches and discussed their impact on the analysis of the phosphoproteome. As pointed out by the authors, similar comparisons have been done for protein levels; however phosphorylation data might behave differently, given the fact that most of the sites can only be determined on the basis of one peptide measurement. Nevertheless, one of the key points of the analysis (the comparison of MS2 vs MS3 modes in isobaric labeling of phosphopeptides) has been studied recently by others (Erickson et al. and Huang et al) which reduces to some extent the novelty of the work. In addition, the authors also proposed a new strategy to determine the stoichiometry of phosphorylation using the high accuracy achieved with MS3, which is quite significant. The manuscript is well written, the experiments are well executed and the data is presented transparently. However, the manuscript deals with a very specific technical issue questioning whether the broad audience of Nat Communications is the right target. Alternatively, the manuscript might better fit on a specialized journal wherein will be well received by the MS community, given the technical importance of the question under study here.

ANSWER: We thank the reviewer for his or her thorough and insightful evaluation of our manuscript. We are happy to learn that this reviewer appreciates the significance of our new strategy to determine phosphorylation site stoichiometry, and that he or she generally finds our manuscript well written and the experiments well executed and presented transparently. For the reviewer's comments about reduced novelty of our work and the choice of target journal we disagree and address both points in details in the paragraphs below named **Point 5: MS2 vs. MS3 comparison novelty** and **Point 6: Audience specificity**.

Some comments to the authors:

- The authors applied the Significance A to be able to analyze the SILAC data. However, in my opinion, the use of this test in this scenario is incorrect for two reasons. i) the biological triplicates are collapsed into a single ratio, thus the reproducibility of the independent measurements is not considered, ii) this is an outlier test and therefore is influenced by the whole distribution of ratios: it fails identifying significant changes when the two samples under comparison are very different. The latter is observed in the re-analysis of Huang et al. data (Suppl. Fig 5) in which two breast cancer cell lines were compared. While the t-test found 40% of the phosphosites as statistically different, the Significance A only found 1.8%. As an alternative to Sig A, the authors might consider the use of one-sample t-test as a mean to compare all four strategies under the same statistical criteria.

ANSWER: We agree with the reviewer that the significance A test is not the best option for identifying significantly regulated phosphorylation sites. We thank the reviewer for suggesting t-testing as a more appropriate alternative and as a means to compare all four strategies under the same statistical criteria. We have followed this suggestion of the reviewer and have reanalyzed all datasets using the well-established Significance Analysis of Microarrays (SAM) statistical test, which combines multiple-testing-corrected t-testing with

background variance estimation. Please see a detailed explanation of the new results in the paragraph below named **Point 1: Significance A-test and fold-change criterion**.

- Figure 4. Why the authors used only the top 1% of Sig A up-regulated sites? Do the conclusions change if the two-sample t-test (at least for LFQ, MS2 and MS3 data) up-regulated sites are used instead?

ANSWER: This is a mistake in the figure legend. We now base the analysis as suggested on the t-test significant sites. Please see a detailed explanation of the new results in the paragraph below named **Point 7: Top 1% regulated sites**.

- Figure 4b. The p-values should be adjusted to q-values (testing the entire GO database can result in a large fraction of false positive enriched terms)

ANSWER: We agree with the reviewer that p-values should be adjusted to q-values when performing GO enrichment analysis, and we now reanalyzed our dataset accordingly. Please see a detailed explanation of the new results in the paragraph below name **Point 8: GO term FDR-adjustment**.

- Line 181: the low precision found on the TMT MS2 data (Fig 2b and 2c) is just a consequence of the nature of the isobaric interference issue, thus it is not strictly correct to state that this strategy yields the highest precision. The variance of these measurements will be inevitably smaller given the significant contribution of the interference (whose variance tends to 0)

ANSWER: We agree that the apparent higher precision of MS²-based TMT is at least partly due to isobaric interference from co-fragmented peptides. However, as this needs to be explored in more detail, we now performed a number of new experiments and statistical analyses, which are described in detail in the paragraph below named **Point 4: TMT precision**.

- Figure 3c. For the same reason as above, the small variance of the MS2 data favors the identification of statistically significant changes in phosphosites whose mean difference is marginal (and therefore biologically irrelevant). The authors should apply, on top of the q-value, a second criterion based on the magnitude of change, i.e. fold-change (see for instance Gai Gianetto et al. Proteomics 2016)

ANSWER: We agree with the reviewer that we need to apply a fold-change in addition to the q-values to overcome the issue with marginally changing sites that are deemed significant. We have followed this suggestion of the reviewer and have reanalyzed all datasets using the well-established Significance Analysis of Microarrays (SAM) statistical test, which combines multiple-testing-corrected t-testing with background variance estimation, and thus gives higher statistical weight to more extreme fold-changes. Please see a detailed explanation of the new results in the paragraph below named **Point 1: Significance A-test and fold-change criterion**.

- Line 210: the authors used a long gradient (5h) for LFQ (to compensate the lack of pre-fractionation in this technique). However, the performance of the chromatography (e.g. peak shapes, peak intensities) may have an impact on MS1-based quantification. Albeit this will reduce the number of identifications, would shorter gradients result in MS1 data of a higher quality and therefore improve the statistical outcome of this approach?

ANSWER: We fully agree with the reviewer on this point and thank him or her for the suggestion of performing LFQ experiments with different LC gradients. Reviewer #1 also pointed out the importance of addressing the influence of more replicates and different LC gradient lengths on the number of significantly regulated sites. To address this in detail, we have therefore performed new label-free phosphoproteomics experiments accordingly. Please see a detailed explanation of the new results in the paragraph below named **Point 3: LFQ gradient length and number of replicates**.

Point 1: Significance A-test and fold-change criterion

Reviewer #2 (Remarks to the Author):

Some comments to the authors:

- The authors applied the Significance A to be able to analyze the SILAC data. However, in my opinion, the use of this test in this scenario is incorrect for two reasons. i) the biological triplicates are collapsed into a single ratio, thus the reproducibility of the independent measurements is not considered, ii) this is an outlier test and therefore is influenced by the whole distribution of ratios: it fails identifying significant changes when the two samples under comparison are very different. The latter is observed in the re-analysis of Huang et al. data (Suppl. Fig 5) in which two breast cancer cell lines were compared. While the t-test found 40% of the phosphosites as statistically different, the Significance A only found 1.8%. As an alternative to Sig A, the authors might consider the use of one-sample t-test as a mean to compare all four strategies under the same statistical criteria.

[...]

- Figure 3c. For the same reason as above, the small variance of the MS2 data favors the identification of statistically significant changes in phosphosites whose mean difference is marginal (and therefore biologically irrelevant). The authors should apply, on top of the q-value, a second criterion based on the magnitude of change, i.e. fold-change (see for instance Gai Gianetto et al. Proteomics 2016)

We thank the reviewer for this suggestion and agree that 1) the significance A-test is not the correct way to test for significant regulation in a data set and 2) that the fold-change should be taken into account in some way. This is why we decided to redo all significant regulation-testing in our manuscript with statistical analysis of microarrays (SAM)-testing, which was cited by the reviewers in their reference, and which is broadly applied in proteomics literature by close to 500 citations of the original manuscript describing the method (Tusher et al, 2001).

We first tried out the one sample t-test approach suggested by the reviewer. However, while this did not significantly increase the performance of SILAC, it essentially negated all significant regulation in both the LFQ and TMT datasets, while TMT still performed slightly better than SILAC (data not shown). We believe that the one sample t-test neglects half of the provided information of LFQ and TMT intensity data by transforming them into ratios, which poses a substantial disadvantage for both LFQ and TMT. In addition, one sample t-testing does not provide an unbiased solution to include fold-changes, as suggested by the reviewer. For this reason, we next considered the suggested paper from Gianetto et al. (Proteomics 2016), which discusses an approach called significance analysis of microarrays or SAM-testing. This approach combines multiple-testing-corrected t-testing with background variance estimation. The original paper from Tusher et al. published 2001 in PNAS has almost 12.000 citations on google scholar, and a variation of the test is implemented in the popular computational proteomics data analysis software Perseus (Tyanova et al. Nat Methods 2016). In an initial test, it proved to work well with all quantification approaches, we had used in our experiments. For SILAC, we found that we can improve the testing results by manually deriving intensities from the MaxQuant-provided normalized ratios and total

triplet intensities, and we thus incorporated this approach into our analysis workflow (see updated Online Methods section).

In our routine data analysis strategies in the lab, we de facto already use SAM testing with the s_0 “fudge factor” in Perseus. However, this factor, as stated in the publication referenced by the reviewer (Gianetto et al. Proteomics 2016), needs to be tuned to each individual data set at hand, and in Perseus it can only manually be set to fixed values. The fudge factor is a way to estimate the background variance of the data set, which in turn depends on its precision. Since our different quantification approaches have vastly different precisions (see Fig 2c), we initially decided against using it to not introduce a statistical bias into our significance analysis. But reading the comment of the reviewer, we tried to reassess this challenge in an unbiased way. We went back to the original publication of the s_0 factor from Tusher et al. (PNAS 2001). The original authors have now released their proposed “significance analysis of microarrays” or SAM test as an R package called “SAMR” (Tibshirani et al. 2011). This package allows to automatically determine s_0 based on the tested data set in question, which enabled us to run the test without a manually set and thus potentially biased s_0 . We thus applied the SAM test as the standard test to determine significant regulation for this manuscript. This meant that big parts of our data analysis, including original Figures 2, 3, 4, S4 and S5 had to be redone. Importantly, even though the total number of significantly regulated sites changed for each method under the conditions described above, our conclusions stay the same, as depicted in the revised main text below.

To assess the biological effect of 2 h doxorubicin treatment on cellular signaling, we next aimed at identifying significantly up- or down-regulated phosphorylation sites. For this purpose, we only used phosphorylation site ratios quantified in all three biological replicates and with an at least 75% probability of correct phosphorylation site localization (Fig. 3b). These strict requirements exclude most falsely quantified or localized sites, which is commonly done in cell signaling studies. We then identified significantly regulated phosphorylation site ratios among the remaining sites via SAM testing (Fig. 3c). Due to its low precision and phosphopeptide coverage, LFQ yielded only 62 significantly regulated phosphorylation sites, and even using nearest neighbour-based imputation of values only slightly increased this number (Fig. 3d). Activating MBR on the other hand multiplied the number of regulated sites more than four times to 279. SILAC performs significantly better with 738 sites deemed significantly regulated, and 908 when activating MBR and REQ. Interestingly, this number doubles when using imputation to account for missing values. Still, the TMT-based methods identify the highest total numbers of significantly regulated sites, with 2140 for the MS^3 -based and 5045 for the MS^2 -based approach. The more than doubled number of hits for MS^2 -based over MS^3 -based TMT seems to be mainly caused by the former’s higher phosphopeptide identifications. In addition, the increase in accuracy of MS^3 -based over MS^2 -based for TMT quantification is essentially negated by its even bigger decrease in precision. This is shown when looking at relative numbers, as MS^2 -based TMT is able to deem a larger fraction of its quantified phosphorylation sites as significantly regulated than the MS^3 -based approach (Fig. 3e). Despite these substantial differences in total numbers, at least 57% of phosphorylation sites tested in all methods were identified as significant in at least one other method as well (Fig. 3f).

Figure 3: Evaluation of quantification methods with focus on identification of significantly regulated phosphorylation sites for doxorubicin vs. control. a) Non- or SILAC-labeled U2OS cells were treated with 5 μM doxorubicin (DOX), 2.5 μM 4-Nitroquinoline 1-oxide (4NQO) or DMSO (C) for 2 h before lysis. Biological triplicates were performed for all quantification methods. For MS measurement, each quantification method was given a total of two days instrument time (including LC overhead), to mimic instrument time-limiting conditions of a comprehensive biological study. This allowed for fractionation of SILAC samples into ten fractions per sample on an Ultimate 3000 high-flow system, and TMT into 24 fractions total on an Ultimate 3000 micro-flow system. Samples were then measured using a 15 or 50 cm (only LFQ) column on a Q Exactive HF or Orbitrap Fusion Lumos (only TMT MS³ OT MC). For SILAC and TMT, MS samples were injected without dilution, so that each labeling channel resembles one LFQ injection. b) Bar plot showing total numbers of identified and quantified phosphopeptides for all replicates of each quantification method, respectively. Calculations of ratios were performed within biological replicates and filtered for measurement in a minimum of one, two or three replicates,

and >75% confident phosphorylation site localization. For further analysis, ratios quantified in all three replicates only and with a localization probability of at least 75% (black arrows) were used. c) SAM-based identification of significantly regulated phosphorylation sites was performed with two sample paired t-test and standard settings (s_0 estimation automatic, delta estimation based on $FDR = 0.20$). Significantly regulated phosphorylation sites (sig) are highlighted in red, non-significant ones in grey. Applied s_0 and delta values, as well as the total number of tested phosphorylation sites (n) are shown. For LFQ and SILAC nearest neighbour imputation (IMP), phosphorylation sites quantified in at least one replicate and with a localization probability of at least 75% were used. Imputation was performed with the R package "impute" incorporated in the "samr" package using standard settings. d/e) The bar plots show the number of significantly regulated phosphorylation sites for each quantification method d) in total, and e) as a fraction relative to the total number of tested sites. f) The Venn diagrams show the overlap of SAM-regulated phosphorylation sites identified in total, and for common identifications between all four shown methods.

The analysis of treatment with 4NQO vs. control yielded overall the same conclusions as doxorubicin (Supplementary Fig. 4a-d), but since 4NQO induces a weaker rewiring of the phosphoproteome, we identified less significantly regulated sites for each of the quantification methods compared to doxorubicin. Importantly, the observed loss in MS^3 -based TMT phosphoproteome coverage and identification of significantly regulated phosphorylation sites compared to MS^2 is not restricted to our dataset, but is also observed when reanalyzing published data. Huang et al. published a data set with complementary MS^2/MS^3 -based TMT quantification of breast cancer cell lines⁴⁷. Comparing the basal phosphoproteome of cell lines AU565 vs. T47D in technical duplicates confirms that while MS^3 -based TMT decompresses MS^2 -measured ratios, the latter allows the identification of more significantly regulated phosphorylation sites (Supplementary Fig. 5).

Supplementary Figure 4: Evaluation of quantification methods with focus on identification of significantly regulated phosphorylation sites for 4NQO vs. control. a) Bar plot showing total numbers of identified and quantified phosphopeptides for all replicates of each quantification method, respectively. Calculations of ratios were performed within biological replicates and filtered for measurement in a minimum of one, two or three replicates, and >75% confident phosphorylation site localization. For further analysis, ratios quantified in all three replicates only and with a localization probability of at least 75% (black arrows) were used. b) Significance analysis of microarrays (SAM)-based identification of significantly regulated phosphorylation sites was performed with two sample unpaired t-test and standard settings (s0 estimation automatic, delta estimation based on FDR = 0.20). Significantly regulated phosphorylation sites (sig) are highlighted in red, non-significant ones in grey. Applied s0 and delta values, as well as the total number of tested phosphorylation sites (n) are shown. For LFO and SILAC nearest neighbour imputation (IMP), phosphorylation sites quantified in at least one replicate and with a localization probability of at least 75% were used. Imputation was performed with the R package “impute” incorporated in the “samr” package using standard settings. c/d) The bar plots show the number of significantly regulated phosphorylation sites for each quantification method c) in total, and d) as a fraction relative to the total number of tested sites. e/f) Heat maps showing e) a kinase motif and f) GO-term enrichment of significantly SAM-up/down-regulated phosphorylation sites from b) vs the respective non-regulated sites as background. Enrichment was performed using Fisher exact tests within Perseus with relative enrichment on gene level and an FDR of 0.02. The numbers above the heatmap show the total number of enriched motifs/GO-terms, while the heat maps below show e) selected motifs or f) all GO-terms with “damage”, “repair”, “checkpoint”, “cell cycle” or “chromosome”, indicative of an activated DDR, respectively.

Supplementary Figure 5: Evaluation of quantification methods on a TMT data set published by Huang et al. a) We reanalyzed a MS²- and MS³-measured TMT data set of different breast cancer cell lines, including AU565 and T47D in technical replicates. In their setup, Huang et al. fractionated peptides after TMT-labeling into twelve fractions on an Agilent 1100 system and subsequently enriched phosphopeptides from each fraction using titanium dioxide. Samples were then measured on an Orbitrap Fusion or Q Exactive HF, for MS³- and MS²-based TMT, respectively. The MS³-setup they used roughly corresponds to the TMT MS³ OT setup as described in this manuscript. Raw files were downloaded from MassIVE and processed as described in the materials and methods section. b) Bar plot showing total numbers of quantified phosphopeptides for all replicates of each quantification method, respectively. Calculations of ratios were performed within biological replicates and filtered for measurement in a minimum of one or two replicates, and >75% confident phosphorylation site localization. For further analysis, ratios quantified in both replicates only and with a localization probability of at least 75% (black arrows) were used. c/d) SAM-based identification of significantly regulated phosphorylation sites was performed with two sample unpaired t-test and s0 estimation set to automatic for the c) TMT MS²- and d) TMT MS³-measured data. Since the delta estimation at standard FDR of 0.20 deemed both data sets almost completely significant, the FDR was decreased to 0.01 to better illustrate the relative differences between both quantification approaches. Significantly regulated phosphorylation sites (sig) are highlighted in red, non-significant ones in grey. Applied s0 and delta values, as well as the total number of tested phosphorylation sites (n) are shown. e/f) The bar plots show the number of significantly regulated phosphorylation sites for each quantification method e) in total, and f) as a fraction relative to the total number of tested sites. g) Correlation of MS²-based and MS³-based TMT quantified AU565 vs. T47D log₂ ratios. The slope of the linear regression line of the SAM-regulated phosphorylation sites is shown and corresponds well to the value determined in this study for TMT MS³ OT as shown in Fig 1 b).

Point 2: LFQ performance and missing values

Reviewer #1 (Remarks to the Author):

The study Benchmarking LFQ, SILAC and MS2/MS3-based TMT quantification strategies for large-scale phosphoproteomics sets out to essentially find the best way to perform quantitative phosphoproteomics experiments as well as generating a new occupancy algorithm. Proteomics researchers can now routinely produce high quality data for, often, nearly the entire proteome. Yet, due to poor overlap between runs and reduced precision, quantitative phosphoproteomics remains more challenging. The work presented here is timely since this is likely the next area that must excel for broader proteomics applicability. The comparisons included in this study are thorough and well designed. I think these types of comparisons are exceptionally difficult to perform because each of these quantitative methods is a skill and constantly changing. Comparing optimized with optimized is the best that can be done but I think the results can be often over stated as what works well for one lab may not for another. That said I think it is an important study and may lead to the field examining phosphoproteomics methods more. The study will certainly be useful as a reference.

I didn't find anything particularly surprising except for how poorly LFQ performed. It's not clear to me how missing values were treated. If a phosphorylation exists in only one of the conditions then the site should really only be measurable in that condition. If you throw these out, then you are throwing out the biggest changers.

[...] I am surprised that more phospho identifications were found with TMT data than the LFQ, even considering the TMT data was fractionated. This doesn't agree with my experience.

I appreciate the authors detailing the different TMT methods they used and what the results were for each. I think this is a big point of confusion for phosphoproteomic work.

We thank the reviewer for his or her comments on the LFQ performance and the missing value problem and the importance of describing this in the manuscript text. We apologize for not going more into detail about how missing values were treated, which we now do. We would like to address both points by illustrating how LFQ missing values were treated in the manuscript at hand, and also why in our experience TMT combined with extensive fractionation can well compete with LFQ in regards of identifications.

First of all, we generally observed that TMT samples generate at least 25% less identifications compared to LFQ on the same LC-MS/MS settings, due to the lower identification rate of TMT. SILAC as well yields less identifications than LFQ due to its higher MS1 spectra complexity. In our technical comparison (Fig. 2), this difference between LFQ vs SILAC/TMT is even more pronounced, as we tried to keep total SILAC/TMT injection intensities equivalent to those of LFQ. The figure below illustrates the total number of peptides quantified on average per MS run and shows that MS²-based TMT quantifies almost half as many peptides as LFQ, and MS³-based TMT yields even less. It's important to note that SILAC is essentially a biological replicate here, complicating a direct comparison. The figure visualizes that under same LC-MS settings, SILAC and TMT can indeed not compete with LFQ in peptide identifications/quantifications.

However in Fig. 3, we fractionate both SILAC and TMT with our routine, and highly optimized high-pH reversed phase fractionation approach (see Batth et al., JPR 2014). Thereagainst, LFK - due to the two day time limitation - is only measured in single shots. Thus, LFK yields a lower coverage of phosphopeptides compared to SILAC and TMT. We repeated the LFK 290 min measurements with new biological samples on our new Q Exactive HF-X instrument (see review point 3) and due to the faster scanning speed of the HF-X, we get more identifications with this new measurement than on the HF. But even with this new LFK data set with broader coverage, we still yield less phosphopeptides compared to the fractionated TMT approach on the HF. In conclusion, we are convinced that an LFK sample fractionated in the same way as TMT would indeed gain a deeper phosphopeptide coverage, but that a single-shot LFK approach can not compete with the coverage of a highly fractionated, TMT-labeled sample.

Regarding the second point, LFK-based methods will of course have many missing values between conditions in standard DDA-based approaches, as indeed observed in this study (see especially Fig. 3b). We think this is a crucial disadvantage of LFK approaches, as a significant part of quantified information cannot be directly used during analysis. Since it is standard in current biological studies to require at least three biological replicates per measurement, we decided in both our technical and biological comparison (Fig. 2 and 3) to only consider ratios quantified in all three out of three replicates and discard the rest. We are aware that, as reviewer #1 mentions, this might lead to a loss of highly regulated sites, but argue that there is no ideal way yet to prevent this. In routine analysis, many users would try to handle this by either using functions such as MBR or applying imputation to rescue a majority of missing LFK values. We now included both methods in our study. When using MBR, one has to accept that localization information cannot be transferred between peptides, and currently no proper FDR-filtering exists to control for false-positive matching. However, accepting these premises as in Fig. 3b, we now see a more than four times increase in significantly SAM-regulated phosphorylation sites from 62 for LFK to 279 for LFK MBR, and these actually add biological information on a systems-level (see Fig. 3c/d and Fig 5a-c). We think that this is a good approach to deal with the LFK missing value problem.

If MBR is not available or applicable, imputation is the second option many users resort to. To address this as well, we newly added nearest neighbor-based imputation to the revised biological benchmark in Fig. 3 assessing the LFK-measured values with SAM testing, which

is implemented in the SAMR R package (see review point 1). This allows testing of phosphosites measured only once in any condition, and boosts the number of tested phosphosites from 3254 when considering >75% localized ratios quantified in all three ratios to 9420 for >75% localized phosphosites in at least one tested condition for LFQ (see Fig. 3b). Importantly however, this did not lead to a significant increase of regulated sites for LFQ (see Fig. 3c/d). And while SILAC did indeed profit substantially from imputation by more than doubling its regulated sites from 738 to 1871, the subsequent biological GO term analysis does not reveal them to be more biologically meaningful on a systems-level (see. Fig 5b/c).

Finally, we noticed that the total number of significantly regulated sites of all quantification approaches, but especially LFQ, greatly depends on the applied normalization strategy and statistical testing conditions. For LFQ, the total number of significantly regulated sites can vary from as low as 0 to as high as 2146. Importantly however, our main conclusion, which is that LFQ performs worse than SILAC, and SILAC in turn worse than TMT, does not change in all the conditions we tested. We created a new supplementary table 3 summarizing these different test results, so that the reader can be made aware of the strong test-dependency (see also revised main text under review point 3).

Supplementary Table 3: Evaluation of quantification methods for doxorubicin vs. control with different statistical approaches. The table lists the total number of tested phosphorylation sites for each quantification approach, and how many of them were deemed statistically significantly regulated by different test settings and normalization approaches. SAM testing was performed using standard parameters (s_0 determination automatic, delta estimation with an FDR cutoff of 0.20). Perseus FDR-corrected t-testing was performed using standard settings (FDR = 0.05), with s_0 set to either to 0.1 (Perseus-default) or 0.2, which were found to be common choices in the literature. In addition to the paired t-test, which was used to correct for day-to-day variance in the biological replicates, a different normalization approach was used together with unpaired t-testing. In this approach, both conditions DOX and C of each biological replicate for each method were normalized by subtracting the average between them in log2 space (= row normalization). Similarly to the paired t-test, this should assure that the statistical test detects biological changes caused by the treatment, and not the day-to-day variation in cell culture and lysing procedure.

Statistical test	LFQ	LFQ MBR	LFQ IMP	SILAC	SILAC MBR REQ	SILAC IMP	TMT MS ²	TMT MS ³
Total sites tested	3254	5143	9420	4453	5098	9420	21,563	13,640
SAMR paired t-test	62	279	738	908	5045	2140	8783	9420
SAMR row-norm. + unpaired t-test	2146	3404	598	3962	49	8862	15,505	9897
Perseus paired t-test $s_0 = 0.1$	0	113	0	2011	2370	2502	2999	1353
Perseus paired t-test $s_0 = 0.2$	0	124	0	1836	2094	2211	2961	1483
Perseus row-norm. + unpaired t-test $s_0 = 0.1$	168	389	20	2721	123	4091	5877	3167
Perseus row-norm. + unpaired t-test $s_0 = 0.2$	155	422	20	2314	117	2997	5180	3126

Point 3: LFQ gradient length and number of replicates

Reviewer #1 (Remarks to the Author):

Also I would have suggested doing more replicates with LFQ with shorter run times. We don't see much improvement after 3 hrs, more replicates can somewhat make up for the missing values.

Reviewer #2 (Remarks to the Author):

- Line 210: the authors used a long gradient (5h) for LFQ (to compensate the lack of pre-fractionation in this technique). However, the performance of the chromatography (e.g. peak shapes, peak intensities) may have an impact on MS1-based quantification. Albeit this will reduce the number of identifications, would shorter gradients result in MS1 data of a higher quality and therefore improve the statistical outcome of this approach?

We thank the reviewers for their comments and agree that there are different factors in our workflow that might have a crucial impact on LFQ numbers of identifications and significantly regulated sites. To test this, we decided to record two new LFQ phosphoproteomics datasets in which we compared four different gradient lengths and up to six replicates. The new analysis illustrates the significant impact of LFQ gradient length and number of replicates for the identification of significantly regulated phosphorylation sites. We can conclude that, while varying the gradient length indeed has an apparently non-systematic impact on the number of significantly regulated sites, the applied 50 cm column 290 min approach seems to be the best single-shot solution for LFQ. Furthermore, while additional replicates indeed substantially improve LFQ quantification, we argue for keeping the number of replicates the same for all tested quantification approaches.

Figure 4: Impact of LFQ gradient time or number of replicates on identification of significantly regulated phosphorylation sites. *a*) U2OS cells were treated with 5 μ M doxorubicin (DOX) or DMSO (C) for 2h before lysis. For the gradient experiment *b-d*), samples were measured in biological triplicates using a 15 cm column with a 30, 90 or 180 min gradient, or a 50 cm column with a 290 min gradient on a Q Exactive HF-X. The shorter gradients are all time-compressed versions of the 290 min gradient, and all other LC-MS instrument settings were kept identical between conditions. For the number of replicates experiment *e-g*), samples were measured in six replicates using the 90 min gradient setup on a Q Exactive HF-X, and three to six biological replicates (replic.) were used for statistical analysis. *b/e*) Bar plots showing total numbers of identified and quantified phosphopeptides for the depicted gradients and number of replicates. Calculations of ratios were performed within biological replicates and filtered for measurement in a minimum of one, two or three replicates, and >75% confident phosphorylation site localization. For further analysis, ratios quantified in all three replicates only and with a localization probability of at least 75% (black arrows) were used. *c/d/f/g*) The bar plots show the number of significantly regulated phosphorylation sites for each quantification method *c/f*) in total, and *d/g*) as a fraction relative to the total number of tested sites. SAM-based identification of significantly regulated phosphorylation sites was performed with two sample paired t-test and standard settings (s0 estimation automatic, delta estimation based on FDR = 0.20).

Regarding gradient length, we would also expect that with the same column length, shorter gradients, while reducing the number of total peptide identifications, should yield more robust

quantification due to enhanced chromatographic performance (sharper peaks) resulting in overall improved ion statistics. We decided to test this in a new experiment based on the setup of our biological benchmark (Fig. 3, 4a). As before, we stimulated U2OS cells for 2 h with 5 μ M doxorubicin vs. DMSO as a control, and then measured three replicates each on a short 15 cm column with a 30, 90 and 180 min gradient, and on the long 50 cm column with the original 290 min gradient used in the biological benchmark in Fig 3. The shorter gradients were compressed versions of the 290 min gradient and the MS settings were kept identical, to ensure a direct technical comparison between the gradients. The number of quantifications can be seen in the new Fig. 4 above. As expected, the number of quantified peptides increases dramatically with longer gradient lengths (Fig. 4b). The increase is almost linear going from 1197 quantified peptides for the 30 min gradient to 3265 for 90 min, but less profound with 4137 for doubling the gradient length again to 180 min. However, switching to a long 50 cm column and a 290 min gradient again yields an almost linear increase to 5966 quantified peptides. This indicates that while long gradients up to 3 h do indeed not add many identifications as pointed out by reviewer #1, switching to a longer column can extend profitable gradient lengths to more than 3 h, presumably by boosting chromatographic performance. It has to be noted that this new data was acquired on a Q Exactive HF-X, as the HF instrument on which we measured the original data is no longer available. Due to the faster scan speed and higher sensitivity of the HFX instrument over the HF (see Kelstrup et al., JPR 2017), the new long 50 cm column 290 min data yields a deeper coverage. Even though this was not tested in a direct instrument-to-instrument comparison, the brighter ion source might enable higher ion intensities, leading to more robust quantification and statistics, and thus not being directly comparable to the original HF data set. However, the relative conclusions from the HF-X data should apply to our biological benchmark recorded on the HF as well.

If we now apply SAM testing to identify significantly regulated phosphopeptides for these different gradient lengths, we do see 207 significantly regulated sites for the short 15 cm column 30 min gradient as compared to only 160 for the 90 min gradient (Fig 4c/d). Surprisingly however, the 180 min gradient with 412 yields significantly more hits than the 30 min gradient. This might indicate that the robustness of the quantification is not only influenced by the actual gradient length, but other factors as well, such as run-to-run LC-MS performance. Importantly however, the number of significantly regulated phosphosites on the long 50 cm column with a 290 min gradient is with 460 the highest of all tested gradients. We speculate that switching from a short 15 cm column to a long 50 cm column results in better chromatographic performance at high gradient lengths. In any case, this finding underlines that choosing the 290 min gradient on the long 50 cm column is our best optimized gradient approach to measure LFQ in the biological benchmark in Fig. 3.

Next, we wanted to assess how a higher number of biological replicates would impact the number of significant hits on the 90 min gradient on our short 15 cm column. As shown in Fig. 4f, this setup on the HF-X yielded only 160 significantly regulated sites. By just adding one additional replicate though, this number increases by a factor of three to 484, which is already slightly more than the 50 cm column 290 min gradient. Including a fifth and sixth replicate again increases the number, but to a lower extent, to 711 and 990, respectively. This demonstrates quite clearly that as the first reviewer suggested, it would be beneficial to

measure more biological replicates with shorter gradient lengths to assure more robust quantification. However, we expect this to hold true for SILAC and TMT as well. Furthermore, using different numbers of replicates for the different quantification approaches would make it very difficult to compare their inherent quantification characteristics with each other. Since most scientific publications use three biological replicates, even if this as our results show is not ideal, we decided to focus our comparison of the quantification approaches in this study on three biological replicates as well.

In conclusion and based on our observations above, we thus argue that measuring LFQ with a long 50cm column on a 290 min gradient and keeping the number of replicates constant between all used quantification approaches, as done in this study (see Fig. 3), is the fairest way to assess the inherent quantification robustness of the tested quantification approaches. At the same time, we would like to include both the experiments on LFQ gradient length and LFQ number of replicates as a new Fig. 5 and point out their importance in this study:

We would like to stress that the actual number of significantly regulated sites identified varies significantly depending on which data normalization approach and statistical test are used (Supplementary Table 3). Especially for LFQ, the number of hits can vary from 0 to 2146. Importantly however, in all the applied tests, the relative conclusions from above do not change, with LFQ always identifying the least number of significantly regulated phosphorylation sites. We were speculating that this could be influenced by the measurement of LFQ in single shots on a long 50 cm column on a very long 290 min gradient. To test this, we repeated the experiment only for LFQ with DOX treatment, but this time varied the gradient length to include 30, 90 and 180 min on a 15 cm column, and our original 290 min 50 cm approach (Fig. 4a). This new data was measured on a Q Exactive HF-X instead of the HF used in our original data set, which lead to slightly better peptide quantifications for LFQ (Fig. 4b/c). This might be influenced by the higher scan speed and brighter ion source of the HF-X, which could lead to higher ion intensities and thus better statistics, even though we have not tested this in a direct comparison 46. Importantly, we still achieve the highest total number of significantly regulated phosphorylation sites with a 290 min 50 cm approach, even though the increase in regulated sites is not correlating well with the high increase in LC-MS time as compared to the very short 30 min gradient (Fig. 4d). We found that a much more efficient approach to boost the numbers of significantly regulated phosphorylation sites for LFQ is to measure more than three replicates (Fig. 4e,f). The number of replicates have a profound impact on the number of significantly regulated sites, which seems to increase linearly as a function of the number of replicates (Fig. 4g). Analyzing four replicates with 90 min runs already results in more significantly regulated sites than three 290 min measurements, while six replicates of 90 min even double the number of significantly regulated sites. For our biological comparison at hand however, we aim to compare the inherent quantification characteristics of the approaches and would further expect similar benefits from more replicates for SILAC and TMT as well. Thus while we encourage users to measure more than three replicates especially for LFQ, we settled to compare SILAC and TMT to the in our hands best performing LFQ approach with 290 min on a 50 cm column with three biological replicates each.

Point 4: TMT precision

Reviewer #1 (Remarks to the Author):

The authors state that the precision of TMT is better when using MS² while noting that this improvement actually comes from precursor interference. This is a common thought that I fundamentally disagree with. The measured precision is assigned to the identified peptide but it's actually a measurement of all the precursors that were co-isolated. The perceived improvement is an artifact and is not necessarily representative of the identified peptide. The phosphorylation model sample is very clever.

Reviewer #2 (Remarks to the Author):

- Line 181: the low precision found on the TMT MS² data (Fig 2b and 2c) is just a consequence of the nature of the isobaric interference issue, thus it is not strictly correct to state that this strategy yields the highest precision. The variance of these measurements will be inevitably smaller given the significant contribution of the interference (whose variance tends to 0)

We thank the reviewers for their comments on MS²-based TMT precision. We agree with the reviewers that the observed higher precision of MS²-based TMT does not necessarily represent a better quantification, which is why we now refer in the text to an increased apparent precision for MS²-based TMT. Furthermore, we also agree that we cannot assume that this increased apparent precision strengthens MS²-based TMT quantification without further testing this. For this purpose, we performed ROC curve analysis on our technical benchmark data (Fig. 2d), which indicates that at least for 10:1 ratios, MS²-based TMT quantification with its higher apparent precision is just as specific in identifying significantly regulated peptides, or even better, than MS³-based approaches.

We wanted to address the problem of assessing MS²-based TMT precision by further exploiting our technical benchmark data (Fig. 2), where all ratio dilutions for all methods were measured in technical triplicates. This should allow us to identify significantly regulated phosphopeptides via SAM-testing. Since we know which peptides should be regulated and which should not due to their yeast/human origin, we argue that this would be a good way to test how reliably the different quantification approaches but especially MS²-based TMT perform. Since a reliable ROC curve analysis required more yeast peptide identifications than present in some LFQ runs in our original data set, we decided to repeat this experiment, while moving the original results to Supplementary Fig. 2b-d. Importantly, the new technical benchmark dataset in Fig. 2 features the same conclusions as before, with slightly lower accuracy for all TMT methods, which we believe to be workflow-related.

The new data set now allowed us to perform SAM analysis for the 4:1 and 10:1 ratios for all quantification approaches, and this time we also included the TMT MS³ IT NL method in our measurement. The SAM test calculates a d-score for each peptide, which reflects its degree of significant regulation. We can now use this d-score and our knowledge that all yeast peptides should be regulated, to calculate true-positive-rates (TPR) and false-positive-rates (FPR). If we plot them against each other, we get a receiver operating characteristic or ROC plot (Fig. 2d). An ideal quantification approach should in this plot reach a 100% TPR before

increasing its FPR over 0%. As expected by both reviewers, we see that for 4:1 ratios, MS²-based TMT performs indeed worse than LFQ, SILAC and most MS³-based TMT methods including TMT MS³ OT MC, which was used in the biological benchmark in Fig. 3. This indicates that at low ratios, the apparent increase in MS²-based TMT quantification precision is indeed an artifact caused by co-isolated peptides. However, when looking at the 10:1 ratios, which as the reviewers highlight in review points 1 and 2 are the biologically more interesting ones, MS²-based TMT performs just as well as LFQ and SILAC, and even slightly better than all TMT MS³ approaches, including OT MC. It is difficult to translate these findings into a general conclusion, as we expect them to depend a lot on the tested ratios, injected MS intensities and the applied statistical test. We highlight this in the text as well. However, we believe it underlines that at least at high ratios, the increased precision of MS²-based TMT seems to add at least partly to its good performance at identifying significantly regulated phosphopeptides and does not seem to entirely be a quantification artifact.

Figure 2: Evaluation of quantification methods with focus on accuracy and precision. a) Yeast phosphopeptides were diluted in fixed ratios 1:4:10 and added to a background of 1:1:1 HeLa phosphopeptides. Same total protein starting amounts were used for each method and SILAC ratios were mixed before digestion. All samples were measured on an Orbitrap Fusion Lumos three times as technical triplicates with each quantification method. For SILAC and TMT, MS samples were diluted to contain a total peptide amount equal to one LFQ injection based on protein starting amount. For TMT, all mixing replicates were measured within the same TMT10-plex run. b) Box plot showing yeast 4:1 and 10:1 phosphopeptide ratios for the different quantification methods and all replicates. Boxes mark the first and third quartile, with the median highlighted as dash, and whiskers marking the minimum/maximum value within 1.5 interquartile range. Outliers are not shown. Both LFQ and SILAC were tested with and without the MaxQuant feature match-between-runs (MBR), and SILAC additionally with both MBR and requantify (REQ) activated. As SILAC-MBR only results were essentially identical to SILAC only, they are not shown here. c) Mean squared errors were calculated as a sum of positive bias and variance for each method and all replicates. d) Receiver operating characteristic (ROC) curves were calculated by using the d-score from SAM testing as an indicator for significant regulation at 4:1 and 10:1 dilution. SAM testing for significantly regulated phosphopeptides was performed at default settings (s0 estimation automatic). ROC plots are presented as zoomed-in excerpts from the total plots, shown on the bottom right each.

To achieve reliable biological interpretation, quantification methods have to be both accurate and precise. To assess this, we first compared how accurate the different quantification methods could measure the 4:1 and 10:1 yeast phosphopeptide ratios (Fig. 2b, Supplementary Fig. 2). LFQ and LFQ MBR both turned out to be the most accurate methods, slightly overestimating ratios on median. SILAC was almost as accurate as LFQ, but underestimated ratios on median. Activating only MBR for SILAC yielded results essentially identical to no MBR (data not shown). However, the quantification accuracy drops significantly when also activating the REQ feature, indicating that this function trading accuracy for an increase in number of quantified sites should be used with caution (Fig. 2b). As expected, MS²-based TMT ratios were heavily affected by ratio compression, resulting in the lowest accuracy of all compared quantification methods. MS³-based TMT methods can rescue the compression to different extents, with the TMT MS³ OT MC method being closest to SILAC. Since the true target ratios were known, we were able to calculate the fraction of reporter ion intensity coming from contaminating ions for MS²-based TMT. We found that it negatively correlated with MS¹ precursor intensity, Andromeda MS/MS score and precursor isolation fraction (PIF) (Supplementary Fig. 3). However, based on the Pearson correlation coefficients none of these turned out to be a robust predictor, which is consistent with what was previously reported for the PIF value²³. We next calculated the mean squared errors (MSE) as the sum of positive bias and variance for each method. These represent the quantification error in accuracy and precision, respectively, and thus allowed for a direct comparison of those two parameters (Fig. 2c, Supplementary Fig. 2c). Of all methods tested, MS²-based TMT yields the highest precision but lowest accuracy. Furthermore, the higher accuracy of MS³-based TMT methods seem to come at the cost of lower precision compared to MS²-based TMT. LFQ, LFQ-MBR and SILAC-MBR-REQ show the lowest precision of all quantification methods. Furthermore, increasing ratios from 4:1 to 10:1 leads to a decrease in precision for all quantification methods.

Next we assessed how the different tradeoffs in accuracy and precision for the quantification approaches would influence their ability to identify phosphopeptides as significantly regulated. This seemed especially interesting for MS²-based TMT, where the apparent increase in quantification precision is easiest explained by the contamination of the TMT reporter ion signal with co-isolated, non-regulated peptides. We would expect that if the gain

in precision is indeed caused solely by a quantification artifact, MS²-based TMT would perform much worse than MS³-based TMT at deeming phosphopeptides significantly regulated. To test this, we applied significance analysis of microarrays (SAM)-testing, which uses t-testing with an added background variance parameter termed s_0 ⁴³. This s_0 parameter gives greater weight to extreme fold changes and should be adjusted to the data set at hand⁴⁴. We used an R package provided by provided by Tibshirani et al.⁴⁵, which can automatically estimate s_0 based on the tested data, and calculates a d-score representing the degree of significant regulation of each tested peptide. Since we know that all yeast peptides should by definition be regulated within our data set, we can use the d-score to calculate true-positive-rates (TPR) and false-positive-rates (FPR) of the up-regulated phosphopeptides for each of the quantification approaches, and plot them in a receiver operating characteristic (ROC) curve (Fig. 2d). In this plot, an ideal quantification approach would reach a TPR of 100%, which means all true positive hits were identified as positive, before the FPR becomes greater than 0%. When looking at the 4:1 ratios, we see that LFQ shows the steepest TPR increase, followed by SILAC and MS³ OT MC-based TMT. As expected, MS²-based TMT performs poorly, indicating that its higher apparent precision is indeed not providing robust quantification of low peptide ratios. At the higher 10:1 ratios however, MS²-based TMT performs equally well as MS³ OT MC-based TMT, and even outperforms it at higher FPRs. We would like to stress that this analysis of course depends on many factors, including chosen ratios, total MS ion intensities and the applied statistical test. It is additionally complicating that the number of identified yeast and human peptides vary between the quantification approaches due to the inherent stochastic behaviour of the data-dependent acquisition (DDA). This is especially true for LFQ/TMT vs SILAC since they are essentially different biological samples (Supplementary Table 2). Nevertheless, we can conclude that especially for the biologically more interesting 10:1 ratios, LFQ, SILAC, MS³ OT MC-based and even ratio-compressed MS²-based TMT all seem to provide good and comparable compromises between sensitivity and specificity.

Point 5: MS2 vs. MS3 comparison novelty

Reviewer #2 (Remarks to the Author):

In their work, Hogrebe et al. compared some of the most popular quantitative MS approaches and discussed their impact on the analysis of the phosphoproteome. As pointed out by the authors, similar comparisons have been done for protein levels; however phosphorylation data might behave differently, given the fact that most of the sites can only be determined on the basis of one peptide measurement. Nevertheless, one of the key points of the analysis (the comparison of MS2 vs MS3 modes in isobaric labeling of phosphopeptides) has been studied recently by others (Erickson et al. and Huang et al) which reduces to some extent the novelty of the work.

We thank the reviewer for pointing out this important studies by Erickson et al. (Anal Chem 2015) and Huang et al. (J Proteome Res 2017). Indeed, we had already cited both publications in our work. More importantly, our TMT MS³ IT NL approach is based on the description of Erickson et al., as stated in the text, and we even use the data set from Huang et al. to strengthen our conclusions of MS²- vs MS³-based TMT performance at identifying significantly regulated sites. However, in our opinion both of these studies focus solely on quantification accuracy, which as we point out in our study is not necessarily the most important quantification characteristic and certainly not the only one to consider. Furthermore, they did not directly compare MS²- vs MS³-based TMT quantification approaches in a biological setting, which we argue highlights nicely the complicated interplay between quantification accuracy and other factors, such as quantification precision and peptide coverage.

Erickson et al show that, as observed in this study as well, MS³-based TMT is able to decompress MS²-quantified TMT ratios and they go on to assess the statistical robustness of MS³-based TMT quantification, but without directly comparing to the MS²-based approach. Huang et al. look at intensity correlations between MS²- and MS³-based TMT, but do not analyze any other quantification or statistical parameters such as precision, sensitivity or specificity. In our study, we aimed at performing all these comparisons in the fairest possible way for all the tested quantification approaches, and we believe our results, especially for the power of MS²-based TMT identification of significantly regulated phosphosites, highlight the necessity of this approach.

Point 6: Audience specificity

Reviewer #2 (Remarks to the Author):

In addition, the authors also proposed a new strategy to determine the stoichiometry of phosphorylation using the high accuracy achieved with MS3, which is quite significant. The manuscript is well written, the experiments are well executed and the data is presented transparently. However, the manuscript deals with a very specific technical issue questioning whether the broad audience of Nat Communications is the right target. Alternatively, the manuscript might better fit on a specialized journal wherein will be well received by the MS community, given the technical importance of the question under study here.

We thank the reviewer for his or her comment and understand that with the level of technical detail we try to address within one publication, this might seem too specific for a broad journal such as Nature Communications. However, we would argue that phosphoproteomics is now such a widely-established field with thousands of publications already, and employed by researchers in many different fields including cell biology, medical research, biotechnology and many others that this should warrant publication in a broad journal such as Nature Communications. Importantly, we aim at giving phosphoproteomics users guidelines for potential advantages, disadvantages and caveats of different quantification approaches. And as choosing LFQ-, SILAC- or TMT-based quantification has to already start at the study design and sample preparation stage, we believe that our conclusions are especially important for scientists from outside the proteomics community, who consider to apply phosphoproteomics to study their own biological questions, and not exclusively interesting for highly specialised proteomics groups. Additionally, many of the analysis strategies used here, such as the ROC plot analysis on our mixed species benchmark to assess sensitivity and specificity of the quantification approaches, and evaluating the impact of technical differences on a biological level, are highly relevant for general omics users as well.

Point 7: Top 1% regulated sites

Reviewer #2 (Remarks to the Author):

- Figure 4. Why the authors used only the top 1% of Sig A up-regulated sites? Do the conclusions change if the two-sample t-test (at least for LFQ, MS2 and MS3 data) up-regulated sites are used instead?

We appreciate this observation made by the reviewer and must admit that we simply forgot to change the Figure legend to reflect what the original Fig. 4 (now 5) actually shows. We deeply apologize for this confusion. We had initially discussed and tested phosphorylation motif analysis with a top 1% regulated criterion of all quantified phosphorylation ratios. This yielded motif plots where the “% difference (p-value = 0.05)” values from the iceLogo test were more similar between the different methods, as shown here:

We assume that a similar relative number of ATM/ATR-regulated [s/t]Q sites out of all quantified sites can be found for each method. Thus, using the top 1% regulated phosphorylation sites for each method yields similar % Difference values in the iceLogo test, as shown above. However, as reviewer #2 mentioned, we also agreed that this arbitrary criterion would be confusing. The purpose of this panel is to show that each method is able to identify the ATM/ATR [s/t]Q motif as enriched within the upregulated sites. And this is also the case when using the SAM-regulated sites, as shown in the new Fig. 5a in the submitted draft (see review point 8). The only thing that had to be adjusted is thus the Fig. 5a legend to reflect that indeed the SAM-regulated sites and not the top 1% were used (see review point 8).

Point 8: GO term FDR-adjustment

Reviewer #2 (Remarks to the Author):

- Figure 4b. The p-values should be adjusted to q-values (testing the entire GO database can result in a large fraction of false positive enriched terms)

We thank the reviewer for this suggestion. In the original R package we had used for GO-term enrichment, the default workflow did not use multiple testing-correction. However, we agree that this is confusing to the reader and not an analysis strategy generally accepted by the proteomics community. In this re-submitted manuscript, we therefore performed GO-term enrichment in the statistical analysis software Perseus with FDR-based multiple testing correction activated (see new Fig. 5b/c below and Supplementary Fig. 4 e,f in review point 1). We used a default FDR of 0.02 and relative enrichment on gene-level (see revised Online Methods). Importantly, this does not change the original conclusions of MS²-based TMT yielding a broader range of more significant GO terms than all other approaches.

Figure 5: Functional characterization of significantly regulated phosphorylation sites. a) iceLogos of the SAM-up-regulated phosphorylation sites from Fig. 4c) vs the respective non-regulated sites as background. The iceLogos show the ATM/ATR kinase substrate [s/t]Q motif significantly enriched for all tested quantification approaches. b/c) Heat maps showing b) a kinase motif and c) GO-term enrichment of significantly SAM-up/down-regulated phosphorylation sites from Fig. 4c) vs the respective non-regulated sites as background. Enrichment was performed using Fisher exact tests within Perseus with relative enrichment on gene level and an FDR of 0.02. The numbers above the heatmap show the total number of enriched motifs/GO-terms, while the heat maps below show b) selected motifs or c) all GO-terms with “damage”, “repair”, “checkpoint”, “cell cycle” or “chromosome”, indicative of an activated DDR, respectively.

After concluding that the four quantification methods can identify different numbers of SAM-regulated phosphorylation sites, we wanted to assess if and to which degree these sites gave us biological insight into the cellular signaling of the doxorubicin-induced DDR. Linear sequence motif analysis of the up-regulated phosphorylation sites revealed that all techniques, including MS²-based TMT, could correctly identify the DDR-induced ATM/ATR kinase substrate motif [s/t]Q as significantly enriched (Fig. 5a)⁴⁸. This is also true when performing linear kinase motif enrichment analysis within Perseus (Fig. 5b). However, when analyzing enriched gene ontology (GO)-terms among the significantly up-regulated phosphorylation site ratios, LFQ was not able to identify any DDR-related terms containing the keywords checkpoint, damage, repair, cell cycle or chromosome (Fig. 5c). Only with MBR was LFQ able to identify terms such as “response to DNA damage stimulus” or “recombinational repair”, which SILAC could with and without MBR REQ. Neither approach however profited from missing value imputation, which like LFQ alone did not yield any significantly enriched GO terms. Importantly, both TMT methods performed a lot better, yielding a broad variety of DDR-related terms. This was not simply due to the broader phosphorylation site coverage of TMT-based approaches, as testing LFQ SAM-regulated sites with the non-regulated MS²-based TMT sites as background did not yield the above-mentioned DDR-terms either (data not shown). The deepest coverage of GO terms with most significant q-values was achieved by the MS²-based TMT method. Importantly, the 4NQO-based DDR phosphorylation landscape yielded the same conclusions (Supplementary Fig. 4e/f).

Further comments

In accordance with the checklist “ncomms_manuscript_checklist.pdf”, we adapted the following in the manuscript:

- “Results of the current study are written in present tense”: the abstract was adapted to comply
- “Divided by subheadings less than 60 characters (incl spaces) that do not contain punctuation”: adjusted lengths of the result subheadings

In addition:

- All Figure references were updated to reflect the newly rearranged and inserted figures
- The material and methods section was updated to reflect the use of Perseus for GO-term enrichment, ggplot for ROC curve plotting, and the R package SAMR for SAM testing; furthermore the data processing and statistical testing sections were expanded to give a better understanding of the performed normalization strategy and applied tests
- To comply with the www.nature.com/authors/policies/data/data-availability-statements-data-citations.pdf, we added the following sentence to the methods section: *Raw MS data files from Huang et al. (2017) were downloaded from MassIVE with identifier MSV000079655.*
- We included new references for the publications on SAM and the newly used HF-X, the and used R packages PREPROCESS, SAMR, IMPUTE

In addition to review points, on page 7

- FROM *For this purpose, we used a mixed species approach in which we mixed phosphopeptides enriched from yeast at fixed 1:4:10 ratios into a 1:1:1 background of HeLa phosphopeptides*
TO *For this purpose, we used a mixed species approach in which we diluted phosphopeptides enriched from yeast at fixed 1:4:10 ratios into a 1:1:1 background of HeLa phosphopeptides*
- FROM *Thus, it should be noted that while all LFQ and TMT samples were essentially the same peptides mixed in different abundances, the SILAC ratios might be influenced by unavoidable biological sample-variation.*
TO *Thus, it should be noted that while all LFQ and TMT samples were essentially the same peptides mixed in different abundances, the SILAC ratios might be influenced by unavoidable biological sample-variation, which we expect to have an impact on assessing its precision.*

In addition to review points, on page 10

- FROM *The three SILAC replica and the TMT samples were fractionated offline by high-pH reversed phase chromatography^{11,12} into ten and 24 fractions, respectively, and analyzed by ~1h LC-MS/MS gradients.*
TO *The three SILAC replica and the TMT samples were fractionated offline by high-pH reversed phase chromatography^{11,12} into ten and 24 fractions, and analyzed by 70 and 90 min LC-MS/MS gradients, respectively.*
- FROM *This significant loss is due to the aforementioned problem of missing values due to the semi-random nature of DDA and can be reduced by activating MBR.*

TO This significant loss is due to the aforementioned problem of missing values caused by the stochastic nature of DDA and can be reduced by activating MBR.

In addition to review points, in discussion on pages 18/19, to reflect review points

- *FROM That is why, even with high ratio compression, MS²-based TMT quantification was able to identify up to twice as many significantly regulated phosphorylation sites than MS³-based TMT methods, both under two variance-dependent cutoffs and standard t-test conditions Our data also shows that this is caused by the higher phosphopeptide coverage of MS²-based TMT, facilitated by its faster peptide scanning speed its higher precision.*

TO That is why, even with high ratio compression, MS²-based TMT quantification was able to identify more than twice as many significantly regulated phosphorylation sites than MS³-based TMT methods based on multiple testing-corrected SAM-testing. Of course, more significant hits do not imply better quantification by themselves. However by demonstrating their meaningful representation of the expected DDR, we argue that they are a good indicator of the quantification performance. Our data also shows that this increase in significant hits is caused by the higher phosphopeptide coverage of MS²-based TMT, facilitated by its faster peptide scanning speed and its higher apparent precision. The higher apparent precision seems to indeed enable robust peptide quantification for MS²-based TMT, as demonstrated by its good compromise of TPR vs FPR in our ROC curve analysis.

- *FROM Our data furthermore indicates that LFQ is the least suitable quantification method for cell signaling studies among the ones we tested, due to its lower precision and missing multiplexing capabilities.*

TO Our data furthermore indicates that LFQ is the least suitable quantification method for cell signaling studies among the ones we tested, due to its lower precision and missing multiplexing capabilities. However, this disadvantage might be counterbalanced by activating MBR or measuring more replicates instead.

- *FROM A direct comparison of MS²- and MS³- based analysis highlighted that the high accuracy of MS³-based TMT quantification is crucial for achieving accurate and reliable stoichiometry information.*

TO A direct comparison of MS²- and MS³- based analysis highlighted that in this context, the high accuracy of MS³-based TMT quantification is crucial for achieving accurate and reliable stoichiometry information.

- *FROM Even though we did not test this with our setup [...]*

TO However, even though we did not test this with our setup [...]

- *ADDED With new developments in LFQ-based data independent acquisition (DIA) quantification ⁴⁶, it might be interesting to see how this approach can compare to TMT multiplexing for quantitative phosphoproteomics experiments, once current DIA limitations such as reliable phosphorylation site localization can be routinely addressed.*

- *FROM To increase phosphopeptide coverage and quantification precision of MS³-based TMT over MS² [...]*

- *TO To increase phosphopeptide coverage and quantification precision of MS³-based over MS²-based TMT [...]*

REVIEWERS' COMMENTS:

Reviewer #1 (Remarks to the Author):

The authors have responded to all my concerns and I am satisfied with the revised manuscript. Actually, the response and revised manuscript is even further improved and should be published as is.

Reviewer #2 (Remarks to the Author):

The authors have now addressed all my concerns in this carefully revised version of the manuscript. The new statistical analysis is quite sound and the conclusions are more robust. Also, they have performed additional experimentation regarding the LFQ strategy which is acknowledgeable. The manuscript should now be accepted.

Point-by-point responses to reviewer's comments

Manuscript number: **NCOMMS-17-21440-T**

Our responses to reviewer comments are provided below in **blue font** after each comment and changes to the manuscript are visualized by *italic blue font*.

Overview reviewer's comments

Reviewer #1 (Remarks to the Author):

The authors have responded to all my concerns and I am satisfied with the revised manuscript. Actually, the response and revised manuscript is even further improved and should be published as is.

ANSWER: Thank you for appreciating the efforts we made in further improving the manuscript, and for suggesting that it is now ready for publication.

Reviewer #2 (Remarks to the Author):

The authors have now addressed all my concerns in this carefully revised version of the manuscript. The new statistical analysis is quite sound and the conclusions are more robust. Also, they have performed additional experimentation regarding the LFQ strategy which is acknowledgeable. The manuscript should now be accepted.

ANSWER: Thank you for appreciating our new experiments and analyses, and for suggesting publication of our manuscript in its current form.